# REWARD-GUIDED FLOW MERGING VIA IMPLICIT DENSITY OPERATORS

## ABSTRACT

Unprecedented progress in large-scale flow and diffusion modeling for scientific discovery recently raised two fundamental challenges: $(i)$ reward-guided adaptation of pre-trained flows, and $(ii)$ integration of multiple models, i.e., model merging. While current approaches address them separately, we introduce a unifying probability-space framework that subsumes both as limit cases, and enables *reward-guided flow merging*. This captures generative optimization tasks requiring information from multiple pre-trained flows, as well as task-aware flow merging (e.g., for maximization of drug-discovery utilities). Our formulation renders possible to express a rich family of *implicit* operators over generative models densities, including intersection (e.g., to enforce safety), union (e.g., to compose diverse models) and interpolation (e.g., for discovery in data-scarce regions). Moreover, it allows to compute complex logic expressions via *generative circuits*. Next, we introduce **R**eward-Guided **F**low **M**erging (RFM), a theory-backed mirror-descent scheme that reduces reward-guided flow merging to a sequential fine-tuning problem that can be tackled via scalable, established methods. Then, we provide first-of-their-kind theoretical guarantees for reward-guided and *pure* flow merging via RFM. Ultimately, we showcase the capabilities of the proposed method on illustrative settings providing visually interpretable insights, and apply our method to high-dimensional de-novo molecular design and low-energy conformer generation.

## 1 INTRODUCTION

Large-scale generative modeling has recently progressed at an unprecedented pace, with flow (Lipman et al., 2022; 2024) and diffusion models (Sohl-Dickstein et al., 2015; Song & Ermon, 2019; Ho et al., 2020) delivering high-fidelity samples in chemistry (Hoogeboom et al., 2022), biology (Corso et al., 2022), and robotics (Chi et al., 2023). However, adoption in real-world applications like scientific discovery led to two fundamental algorithmic challenges: $(i)$ reward-guided fine-tuning, i.e., adapting pre-trained models to maximize downstream utilities (e.g., binding affinity) (e.g., Domingo-Enrich et al., 2024; Uehara et al., 2024b; De Santi et al., 2025b), and $(ii)$ model merging - integrating multiple pre-trained models (Song et al., 2023; Ma et al., 2025), e.g., to incorporate safety constraints (Dai et al., 2023), or unify diverse priors (Ma et al., 2025). The former now benefits from principled and scalable control theoretic or reinforcement learning (RL) methods, with successes in image generation (Domingo-Enrich et al., 2024), molecular design (Uehara et al., 2024b), and protein engineering (Uehara et al., 2024b). By contrast, current merging approaches remain mostly heuristic, training-heavy, and act in weight-space with limited interpretability of the merging operations (Ma et al., 2025; Song et al., 2023). Crucially, these two problems have been treated via distinct formulations and methods. On the contrary, in this work we ask:

*Can we fine-tune a pre-trained flow model to optimize a given reward function while integrating information from (i.e., merge) multiple pre-trained flows?*

Answering this would contribute to the algorithmic-theoretical foundations of *flow adaptation* and enable rich applications in highly relevant areas such as scientific discovery and generative design.

**Our approach** To address this challenge, we first introduce a probability-space optimization framework (see Fig. 1b) that recovers reward-guided fine-tuning and *pure* model merging as limit cases, and provably enables *reward-guided model merging* (Sec. 3). Our formulation allows to express a rich family of *implicit* operators over generative models that cover practical needs such as enforcing safety (e.g., via intersection), composing diverse models (e.g., via union), and discovery

in data-scarce regions (e.g., via interpolation). However, these operators are expressed via non-linear functionals that cannot be optimized via classic RL or control schemes, as shown by De Santi et al. (2025b). To overcome this challenge, we introduce **R**eward-**G**uided **F**low **M**erging (RFM), a mirror descent (MD) (Nemirovskij & Yudin, 1983) scheme that solves reward-guided and pure flow merging via a sequential adaptation process implementable via established fine-tuning methods (e.g., Domingo-Enrich et al., 2024; Uehara et al., 2024b) (Sec. 4). Next, we extend the algorithm proposed, to operate on the space of entire flow processes, enabling scalable and stable computation of the intersection operator (Sec. 5). We provide a rigorous convergence analysis of RFM, yielding first-of-its-kind theoretical guarantees for reward-guided and pure flow merging (Sec. 6). Ultimately, we showcase our method's capabilities on illustrative settings, as well as on a molecular design task for control and optimization of quantum-mechanical properties and conformer generation (Sec. 7).

**Our contributions**   To sum up, in this work we contribute

- A formalization of *reward-guided flow merging* via *implicit operators*, which generalizes recent reward-guided fine-tuning and pure flow merging formulations via an operator viewpoint (Sec. 3).
- *Reward-Guided Flow Merging (*RFM*)*, a principled algorithm which provably solves arbitrary reward-guided flow merging problems via probability-space optimization over the space of data-level marginal densities induced by flow models (Sec. 4), and a stability-enhancing extension for flow intersection following a mirror-descent scheme on the space of joint flow processes (Sec. 5).
- A theoretical analysis of the presented algorithms providing convergence guarantees both under simplified and realistic assumptions leveraging recent understanding of mirror flows (Sec. 6).
- An experimental evaluation of RFM showcasing its practical relevance on both synthetic, yet illustrative settings and on a scientific discovery task, showing it can effectively intersect pre-trained flow models for molecular conformers generation. (Sec. 7).

## 2   BACKGROUND AND NOTATION

**General Notation.** We denote with $\mathcal{X} \subseteq \mathbb{R}^d$ an arbitrary set. Then, we indicate the set of Borel probability measures on $\mathcal{X}$ with $\boldsymbol{P}(\mathcal{X})$, and the set of functionals over $\boldsymbol{P}(\mathcal{X})$ as $\boldsymbol{F}(\mathcal{X})$.

**Generative Flow Models.** Generative models aim to approximately sample novel data points from a data distribution $p_{data}$. Flow models tackle this problem by transforming samples $X_0 = x_0$ from a source distribution $p_0$ into samples $X_1 = x_1$ from the target distribution $p_{data}$ (Lipman et al., 2024; Farebrother et al., 2025). Formally, a *flow* is a time-dependent map $\psi : [0,1] \times \mathbb{R}^d \to \mathbb{R}$ such that $\psi : (t, x) \to \psi_t(x)$. A *generative flow model* is a continuous-time Markov process $\{X_t\}_{0 \le t \le 1}$ obtained by applying a flow $\psi_t$ to $X_0 \sim p_0$ as $X_t = \psi_t(X_0)$, $t \in [0,1]$, such that $X_1 = \psi_1(X_0) \sim p_{data}$. In particular, the flow $\psi$ can be defined by a *velocity field* $u : [0,1] \times \mathbb{R}^d \to \mathbb{R}^d$, which is a vector field related to $\psi$ via the following ordinary differential equation (ODE), typically referred to as *flow ODE*:

$$\frac{\mathrm{d}}{\mathrm{d}t} \psi_t(x) = u_t(\psi_t(x)) \tag{1}$$

with initial condition $\psi_0(x) = 0$. A flow model $X_t = \psi_t(X_0)$ induces a probability path of *marginal densities* $p = \{p_t\}_{0 \le t \le 1}$ such that at time $t$ we have that $X_t \sim p_t$. We denote by $p^u$ the probability path of marginal densities induced by the velocity field $u$. Flow matching (FM) (Lipman et al., 2024) can estimate a velocity field $u^\theta$ s.t. the induced marginal densities $p^{u_\theta}$ satisfy $p_0^{u_\theta} = p_0$ and $p_1^{u_\theta} = p_{data}$, where $p_0$ denotes the source distribution, and $p_{data}$ the target data distribution. Typically FM are rendered tractable by defining $p_t^u$ as the marginal of a conditional density $p_t^u(\cdot|x_0, x_1)$, e.g.,:

$$X_t \mid X_0, X_1 = \kappa_t X_0 + \omega_t X_1 \tag{2}$$

where $\kappa_0 = \omega_1 = 1$ and $\kappa_1 = \omega_0 = 0$ (e.g. $\kappa_t = 1 - t$ and $\omega_t = t$). Then $u^\theta$ can be learned by regressing onto the conditional velocity field $u(\cdot|x_1)$ (Lipman et al., 2022). As diffusion models (Song & Ermon, 2019) (DMs) admit an equivalent ODE formulation with identical marginal densities (Lipman et al., 2024, Ch. 10), our contributions extend directly to DMs.

**Continuous-time Reinforcement Learning.** We formulate finite-horizon continuous-time RL as a specific class of optimal control problems (Wang et al., 2020; Jia & Zhou, 2022; Treven et al., 2023; Zhao et al., 2024). Given a state space $\mathcal{X}$ and an action space $\mathcal{A}$, we consider the transition dynamics governed by the following ODE:

$$\frac{\mathrm{d}}{\mathrm{d}t} \psi_t(x) = a_t(\psi_t(x)) \tag{3}$$

where $a_t \in \mathcal{A}$ is a selected action. We consider a state space $\mathcal{X} := \mathbb{R}^d \times [0,1]$, and denote by (Marko-vian) deterministic policy a function $\pi_t(X_t) := \pi(X_t, t) \in \mathcal{A}$ mapping a state $(x, t) \in \mathcal{X}$ to an action $a \in \mathcal{A}$ such that $a_t = \pi(X_t, t)$, and denote with $p_t^\pi$ the marginal density at time $t$ induced by policy $\pi$.

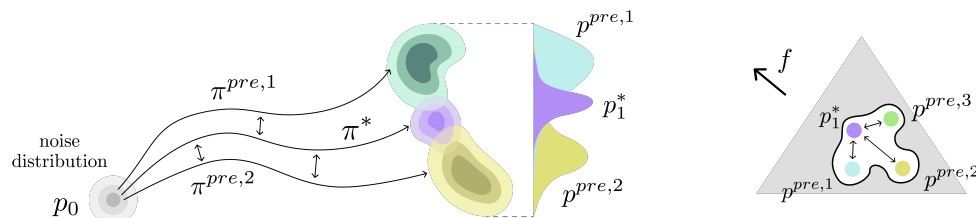

(a) Reward-Guided Flow Merging      (b) Probability-Space Opt. Viewpoint

Figure 1: (1a) Pre-trained and fine-tuned policies inducing $\{p_1^{pre,i}\}_{i=1}^n$ and opt. density $p_1^*$ via reward-guided flow merging. (1b) Probability-space optimization viewpoint on reward-guided merging.

**Pre-trained Flow Models as an RL policy.** A pre-trained flow model with velocity field $u^{pre}$ can be interpreted as an action process $a_t^{pre} := u^{pre}(X_t, t)$, where $a_t^{pre}$ is determined by a continuous-time RL policy via $a_t^{pre} = \pi^{pre}(X_t, t)$ (De Santi et al., 2025a). Therefore, we can express the flow ODE induced by a pre-trained flow model by replacing $a_t$ with $a^{pre}$ in Eq. equation 3, and denote the pre-trained model by its policy $\pi^{pre}$, which induces a density $p_1^{pre} := p_1^{\pi^{pre}}$ approximating $p_{data}$.

## 3 Reward-Guided Flow Merging via Implicit Density Operators

In this section, we introduce the general problem of *reward-guided flow merging* via *implicit density operators*. Formally, we wish to implement an operator $\mathcal{O}: \Pi \times \ldots \times \Pi \to \Pi$ that, given pre-trained generative flow models $\{\pi^{pre,i}\}_{i \in [n]}$, returns a merged flow $\pi^*$ inducing an ODE:

$$\frac{\mathrm{d}}{\mathrm{d}t}\psi_t(x) = a_t^*(\psi_t(x)) \quad \text{with} \quad a_t^* = \pi^*(x, t), \tag{4}$$

such that it controllably merges prior information within the $n$ pre-trained generative models, while potentially steering its density $p_1^* := p_1^{\pi^*}$ towards a high-reward region according to a given scalar reward function $f(x) : \mathcal{X} \to \mathbb{R}$. We tackle this problem by fine-tuning an initial flow $\pi^{init} \in \{\pi^{pre,i}\}_{i \in [n]}$ according to the following optimization formulation, visually portrayed in Fig. 1b.

> **Reward-Guided Flow Merging via Implicit Density Operators**
>
> $$\mathcal{O}: (\pi^{pre,1}, \ldots, \pi^{pre,n}) \to \pi^* \text{ s.t. } \pi^* \in \underset{\pi : p_0^* = p_0^{pre}}{\arg\max} \; \underset{x \sim p_1^\pi}{\mathbb{E}}\left[f(x)\right] - \sum_{i=1}^n \alpha_i \mathcal{D}_i(p_1^\pi \,\|\, p_1^{pre,i}) \tag{5}$$

Here, each $D_i$ is an arbitrary divergence, $\alpha_i > 0$ are model-specific weights, and $p_0^\pi = p_0^{pre}$ enforces that the marginal density at $t = 0$ must match the pre-trained model marginal. This formulation recovers reward-guided fine-tuning (e.g., Domingo-Enrich et al., 2024) when $n = 1$ and $\mathcal{D}_1 = D_{KL}$, and provides a formal framework for *pure* flow merging (e.g., Poole et al., 2022; Song et al., 2023) with interpretable objectives, when the reward $f$ is constant (e.g., $f(x) = 0 \; \forall x \in \mathcal{X}$). In this case, Eq. 5 formalizes flow merging as computing a flow $\pi^*$ that minimizes a weighted sum of divergences to the priors $\{\pi^{pre,i}\}_{i \in [n]}$. Varying the divergences $\{D_i\}_{i \in [n]}$ yields different merging strategies.

**In-Distribution Flow Merging.** Given pre-trained flow models $\{\pi^{pre,i}\}_{i \in [n]}$, we denote by *in-distribution* merging when the merged model generates samples from regions with sufficient prior density. Practically relevant instances include the *intersection operator* $\mathcal{O}_\wedge$ (i.e., a logical AND), and the *union operator* $\mathcal{O}_\vee$ (i.e., a logical OR). Formally, these operators can be defined via:

> **$\mathcal{O}_\wedge$: Intersection ($\wedge$) Operator**
>
> $$\pi^* \in \underset{\pi : p_0^* = p_0^{pre}}{\arg\min} \sum_{i=1}^n \alpha_i \, D_{KL}(p_1^\pi \| p_1^{pre,i}) \tag{6}$$

> **$\mathcal{O}_\vee$: Union ($\vee$) Operator**
>
> $$\pi^* \in \underset{\pi : p_0^* = p_0^{pre}}{\arg\min} \sum_{i=1}^n \alpha_i \, D_{KL}^R(p_1^\pi \| p_1^{pre,i}) \tag{7}$$

The $D_{KL}$ divergences in Eq. 6 heavily penalize density allocation in any region with low prior density for any model $\pi^{pre,i}$, leading to an optimal flow model $\pi^*$ inducing $p_1^*(x) \propto \prod_{i=1}^n p_1^{pre,i}(x)^{\alpha_i}$ (cf. Heskes, 1997). Similarly, the reverse KL divergence $D_{KL}^R(p\|q) := D_{KL}(q\|p)$ in Eq. 7 induces a mode-covering behaviour implying a flow model $\pi^*$ with density $p_1^* \propto \sum_{i=1}^n \alpha_i p_1^{pre,i}(x)$ (cf. Banerjee et al., 2005) sufficiently covering all regions with enough prior density, for any $p_1^{pre,i}$, $i \in [n]$.

**Out-of-Distribution Flow Merging.** We denote by *out-of-distribution*, the case where $\pi^*$ samples from regions insufficiently covered by all priors. An example is the *interpolation operator* $\mathcal{O}_{W_p}$ (see Eq. 8), which induces $p_1^*$ equal to the prior densities Wasserstein Barycenter (Cuturi & Doucet, 2014).

### $\mathcal{O}_{W_p}$: Interpolation (Wasserstein-$p$ Barycenter) Operator

$$\arg\min_{\pi} \sum_{i=1}^{n} \alpha_i W_p(p_1^\pi \,\|\, p_1^{pre,i}) := \sum_{i=1}^{n} \alpha_i \inf_{\gamma \in \Gamma(p_1^\pi, p_1^{pre})} \mathbb{E}_{(x,y) \sim \gamma}[d(x,y)^p]^{\frac{1}{p}} \qquad (8)$$

**Straightforward Generalizations.** While we presented a few practically relevant operators, the framework in Eqs. 5 is not tied to them: it trivially admits any new operator defined via other divergences (e.g., MMD, Rényi, Jensen–Shannon), and allows diverse $D_i$ for each prior flow models $\pi^{pre,i}$. Moreover, sequential composition of these operators makes it possible to implement arbitrarily complex logical operations over generative models. For instance, as later shown in Sec. 7, one can obtain $\pi^* = (\pi^{pre,1} \vee \pi^{pre,2}) \wedge \pi^{pre,3}$ by first computing $\pi_{1,2} := \mathcal{O}_\vee(\pi^{pre,1}, \pi^{pre,2})$ and then $\pi^* := \mathcal{O}_\wedge(\pi_{1,2}, \pi^{pre,3})$. We denote such operators by *generative circuits*, and illustrate one in Fig. 3d.

While being of high practical relevance, the presented framework entails optimizing non-linear distributional utilities (see Eq. 5) beyond the reach of standard RL or control schemes, as shown by De Santi et al. (2025b). In the next section, we show how to reduce the introduced problem to sequential fine-tuning for maximization of rewards automatically determined by the choice of operator $\mathcal{O}$.

## 4 ALGORITHM: REWARD-GUIDED FLOW MERGING

In this section, we introduce **R**eward-**G**uided **F**low **M**erging (RFM), see Alg. 1, which provably solves Problem 5. RFM implements general operators $\mathcal{O}$ (see Sec. 3) by solving the following problem:

### Reward-Guided Flow Merging as Probability-Space Optimization

$$p_1^{\pi^*} \in \arg\max_{p_1^\pi} \ \mathcal{G}(p_1^\pi) \quad \text{with} \quad \mathcal{G}(p_1^\pi) := \mathbb{E}_{x \sim p_1^\pi}[f(x)] - \sum_{i=1}^{n} \alpha_i \mathcal{D}_i(p_1^\pi \,\|\, p_1^{pre,i}) \qquad (9)$$

Given an initial flow model $\pi^{init} \in \{\pi^{pre,i}\}_{i \in [n]}$, RFM follows a mirror descent (MD) scheme (Nemirovskij & Yudin, 1983) for $K$ iterations by sequentially fine-tuning $\pi^{init}$ to maximize surrogate rewards $g_k$ determined by the chosen operator, i.e., $\mathcal{G}$. To understand how RFM computes the surrogate rewards $\{g_k\}_{k=1}^{K}$ guiding the optimization process in Eq. 9, we first recall the notion of first variation of $\mathcal{G}$ over a space of probability measures (cf. Hsieh et al., 2019). A functional $\mathcal{G} \in \boldsymbol{F}(\mathcal{X})$ has a first variation at $\mu \in \boldsymbol{P}(\mathcal{X})$ if there exists a function $\delta\mathcal{G}(\mu) \in \boldsymbol{F}(\mathcal{X})$ such that:

$$\mathcal{G}(\mu + \epsilon\mu') = \mathcal{G}(\mu) + \epsilon\langle\mu', \delta\mathcal{G}(\mu)\rangle + o(\epsilon).$$

holds for all $\mu' \in \boldsymbol{P}(\mathcal{X})$, where the inner product is an expectation. At iteration $k \in [K]$, given the current generative model $\pi^{k-1}$, RFM fine-tunes it according to the following standard entropy-regularized control or RL problem, solvable via any established method (e.g., Domingo-Enrich et al., 2024)

$$\arg\max_{\pi} \ \langle\delta\mathcal{G}\left(p_1^{\pi^{k-1}}\right), p_1^\pi\rangle - \frac{1}{\gamma_k}D_{KL}(p_1^\pi \,\|\, p_1^{\pi^{k-1}}) \qquad (10)$$

Thus, we introduce a surrogate reward function $g_k : \mathcal{X} \to \mathbb{R}$ defined for all $x \in \mathcal{X}$ such that:

$$g_k(x) := \delta\mathcal{G}\left(p_1^{\pi^{k-1}}\right)(x) \quad \text{and} \quad \mathbb{E}_{x \sim p_1^\pi}[g_k(x)] = \langle\delta\mathcal{G}\left(p_1^{\pi^{k-1}}\right), p_1^\pi\rangle \qquad (11)$$

We now present **R**eward-**G**uided **F**low **M**erging (RFM), see Alg. 1. At each iteration $k \in [K]$, RFM estimates the gradient of the first variation at the previous policy $\pi_{k-1}$, i.e., $\nabla_x \delta\mathcal{G}(p_1^{\pi^{k-1}})$ (line 4). Then, it updates the flow model $\pi_k$ by solving the reward-guided fine-tuning problem in Eq. 10 by employing $\nabla_x g_k := \nabla_x \delta\mathcal{G}(p_1^{\pi^{k-1}})$ as reward function gradient (line 5). Ultimately, RFM returns a final policy $\pi := \pi_K$. We report a detailed implementation of REWARDGUIDEDFINETUNINGSOLVER in Apx. E.2.

**Implementation of Intersection, Union, and Interpolation operators.** In the following, we present the specific expressions of $\nabla_x \delta\mathcal{G}(p_1^\pi)$ for pure model merging with the intersection ($\mathcal{O}_\wedge$), union ($\mathcal{O}_\vee$), and interpolation ($\mathcal{O}_{W_p}$) operators introduced in Sec. 3.

$$\nabla_x \delta\mathcal{G}(p_1^\pi)(x) = \begin{cases} -\sum_{i=1}^{n} \alpha_i s^{k-1}(x, t=1) + \sum_{i=1}^{n} \alpha_i s^{\pi^{pre,i}}(x, t=1) & \text{Intersection } (\mathcal{O}_\wedge) \\ -\sum_{i=1}^{n} \nabla_x \exp\left(\phi_i^*(x) - 1\right), \phi_i^* \text{ as by Eq. 45} & \text{Union } (\mathcal{O}_\vee) \\ -\sum_{i=1}^{n} \nabla_x \phi_i^*(x), \phi_i^* = \arg\max_{\phi:\|\nabla_x\phi\|\leq 1}\langle\phi, p^\pi - p^{pre,i}\rangle & \text{Interpol. } (\mathcal{O}_{W_1}) \end{cases}$$

---

**Algorithm 1** **R**eward-Guided **F**low **M**erging (RFM)

---

1: **input:** $\{\pi^{pre,i}\}_{i\in[n]}$ : pre-trained flows, $\{\mathcal{D}_i\}_{i\in[n]}$ : arbitrary divergences, $f$ : reward, $\{\alpha_i\}_{i\in[n]}$ : weighs, $K$ : iterations number, $\{\gamma_k\}_{k=1}^{K}$ stepsizes, $\pi^{init} \in \{\pi^{pre,i}\}_{i\in[n]}$ : initial flow model

2: **Init:** $\pi_0 := \pi^{init}$

3: **for** $k = 1, 2, \ldots, K$ **do**

4:     Estimate $\nabla_x g_k = \nabla_x \delta\mathcal{G}(p_1^{\pi^{k-1}})$ with:

$$
\mathcal{G}\left(p_1^{\pi^{k-1}}\right) = 
\begin{cases}
\displaystyle \mathbb{E}_{x \sim p_1^{\pi^{k-1}}}[f(x)] - \sum_{i=1}^{n} \alpha_i \mathcal{D}_i(p_1^{\pi^{k-1}} \,\|\, p_1^{pre,i}) & \text{(Reward-Guided Flow Merging)} \\[2em]
\displaystyle -\sum_{i=1}^{n} \alpha_i \mathcal{D}_i(p_1^{\pi^{k-1}} \,\|\, p_1^{pre,i}) & \text{(Flow Merging)}
\end{cases}
\tag{12}
$$

5:     Compute $\pi_k$ via standard reward-guided fine-tuning (e.g., Domingo-Enrich et al., 2024):

$$
\pi_k \leftarrow \text{REWARDGUIDEDFINETUNINGSOLVER}(\nabla_x g_k, \gamma_k, \pi_{k-1})
$$

6: **end for**

7: **output:** policy $\pi := \pi_K$

---

Where by $s^{k-1}(x,t) := \nabla \log p_t^{\pi^{-1}}(x)$ we denote the score of model $\pi^{k-1}$ at point $x$ and time $t$, and $s^{pre,i} := s^{\pi^{pre,i}}$. For diffusion models, a learned neural score network is typically available; for flows, the score follows from a linear transformation of $\pi(X_t, t)$ (e.g., Domingo-Enrich et al., 2024, Eq. 8):

$$
s_t^{\pi}(x) = \frac{1}{\kappa_t(\frac{\dot{\omega}_t}{\omega_t}\kappa_t - \dot{\kappa}_t)}\left(\pi(x,t) - \frac{\dot{\omega}_t}{\omega_t}x\right)
\tag{13}
$$

For the union operator, gradients are defined via critics $\{\phi_i^*\}_{i=1}^n$ learned with the standard variational form of reverse KL, as in f-GAN training of neural samplers (Nowozin et al., 2016). For $W_1$ interpolation, each $\phi_i^*$ plays the role of a Wasserstein-GAN discriminator with established learning procedures (Arjovsky et al., 2017). In both cases, each critic compares the fine-tuned density to a prior density $p_1^{pre,i}$, seemingly requiring one critic per prior. We prove that, surprisingly, this is unnecessary for the union operator, and conjecture that analogous results hold for other divergences.

**Proposition 1** (Union operator via Pre-trained Mixture Density Representation). *Given* $\overline{p}_1^{pre} = \sum_{i=1}^n \alpha_i p_1^{pre,i} / \sum_{i=1}^n \alpha_i$, *i.e., the $\alpha$-weighted mixture density of pre-trained models, the following hold:*

$$
\pi^* \in \arg\min_{\pi} \sum_{i=1}^{n} \alpha_i D_{KL}^R(p_1^{\pi} \,\|\, p_1^{pre,i}) = \left(\sum_{i=1}^{n} \alpha_i\right) D_{KL}^R(p_1^{\pi} \,\|\, \overline{p}_1^{pre})
\tag{14}
$$

Prop. 1, which is proved in Apx. D implies that the union operator in Eq. 7 over $n$ prior models can be implemented by learning a single critic $\phi^*$, as shown in Sec. 7. In Apx. C.2, we report the gradient expressions above, and present a brief tutorial to derive the first variations for any new operator.

Crucially, the score in Eq. 13 for the intersection gradient diverges at $t = 1$ ($\kappa_1 = 0$). While prior works attenuate the issue by evaluating the score at $1 - \epsilon$ (De Santi et al., 2025a), this trick hardly scales well to high-dimensional settings. In the following, we propose a principled solution to this problem by leveraging weighted score estimates along the entire noised flow process, i.e., $t \in [0, 1]$.

## 5 TRULY SCALABLE INTERSECTION VIA FLOW PROCESS OPTIMIZATION

Towards tackling the aforementioned issue, we lift the problem in Eq. 6 from the probability space associated to the last time-step marginal $p_1^{\pi}$, where the score diverges, to the entire flow process:

**Intersection Operator $\mathcal{O}_{\wedge}$ via Flow Process Optimization**

$$
\pi^* \in \arg\max_{\pi: p_0^{\pi} = p_0^{pre}} \mathcal{L}_{\wedge}(\mathbf{Q}^{\pi}) := \int_0^1 \lambda_t \sum_{i=1}^{n} \alpha_i D_{KL}(p_t^{\pi} \,\|\, p_t^{pre,i})\,\mathrm{d}t
\tag{15}
$$

Here, $\mathbf{Q}^{\pi} = \{p_t^{\pi}\}_{t\in[0,1]}$ denotes the entire joint flow process induced by policy $\pi$ over $\mathcal{X}^{[0,1]}$. Under general regularity assumptions, an optimal policy $\pi^*$ for Problem 15 is optimal also w.r.t. Eq. 6. Interestingly, an optimal flow $\pi^*$ for Problem 15 can be computed via a MD scheme acting over the space of joint flow processes $\mathbf{Q}^{\pi} = \{p_t^{\pi}\}_{t\in[0,1]}$ determined by the following update rule:

**Reward-Guided Flow Merging (Mirror Descent) Step**

$$\mathbf{Q}^k \in \underset{q:p_0=p_0^{k-1}}{\arg\max} \langle \delta\mathcal{L}_\wedge(\mathbf{Q}^{k-1}), \mathbf{Q}\rangle + \frac{1}{\gamma^k} D_{KL}\left(\mathbf{Q}\|\mathbf{Q}^{k-1}\right) \tag{16}$$

First, we state the following Lemma 5.1, which allows to express the first variation of $\mathcal{L}_\wedge$ w.r.t. the entire flow process $\mathbf{Q}^\pi$ as an integral of first variations w.r.t. the marginal densities $p_t^\pi$.

**Lemma 5.1** (First Variation of Flow Process Functional)**.** *For objective $\mathcal{L}_\wedge$ in Eq. 15 it holds:*

$$\langle \delta\mathcal{L}_\wedge(\mathbf{Q}^k), q\rangle = \int_0^1 \lambda_t \; \mathbb{E}_{\mathbf{Q}}\left[\delta \sum_{i=1}^n \alpha_i \, D_{KL}(p_t^\pi \| p_t^{pre,i})\right] \mathrm{d}t. \tag{17}$$

This factorization of $\langle \delta\mathcal{L}_\wedge(\mathbf{Q}^k), q\rangle$ shows that a flow $\pi_{k+1}$ inducing an optimal process $\mathbf{Q}^k$ w.r.t. the update step in Eq. 16 can be computed by solving a control-affine optimal control problem via the same REWARDGUIDEDFINETUNINGSOLVER oracle used in Alg. 1, by introducing the running cost term:

$$f_t(x) := \delta\left(\sum_{i=1}^n \alpha_i \, D_{KL}(p_t^\pi \| p_t^{pre,i})\right)(x,t), \quad t \in [0,1) \tag{18}$$

This algorithmic idea, which allows to control the score scale at $t \to 1$ via $\lambda_t$, thus enhancing RFM, trivially extends to reward-guided merging, and is accompanied by a detailed pseudocode in Apx. E.2.

## 6 GUARANTEES FOR REWARD-GUIDED FLOW MERGING

In this section, we aim to establish rigorous theoretical guarantees for RFM, ensuring its reliability.

**Central Challenge.** Score functions $s^\pi$ leveraged in Sec. 4 to express gradients of first variations are readily available for pretrained models used to initialize RFM. It is far less clear whether they remain accessible throughout subsequent iterations. In particular, the process returned by REWARDGUIDEDFINETUNINGSOLVER is in general unrelated to the score.

**Score Retention via Stochastic Optimal Control.** Our key observation is that, under a standard approximation, most fine-tuning schemes retain score information. Specifically, we consider fine-tuning through the lens of *stochastic optimal control* (SOC) (cf. Bellman, 1954)), which encompassing many existing methods including Adjoint Matching (Domingo-Enrich et al., 2024), which we employ in Sec. 7. Formally, SOC addresses the following problem defined over SDEs (see Appendix B):

$$\min_{u \in \mathcal{U}} \mathbb{E}\left[\int_0^1 \tfrac{1}{2}\|u(X_t^u, t)\|^2 \, dt - g(X_1^u)\right] \quad \text{s.t.} \quad \mathrm{d}X_t^u = \left(b(X_t^u, t) + \sigma(t)u(X_t^u, t)\right)\mathrm{d}t + \sigma(t)\,\mathrm{d}B_t \tag{19}$$

where $X_0^u \sim p_0$,, $\mathcal{U}$ is the set of admissible controls, and $g$ is a terminal reward, corresponding the $g_k$'s in Algorithm 1. The corresponding *uncontrolled* dynamics (up to a minus sign),

$$\mathrm{d}X_t^u = -b(X_t^u, t)\,\mathrm{d}t + \sigma(t)\,\mathrm{d}B_t, \tag{20}$$

coincide with the *forward process* in diffusion-modeling (Song et al., 2020). We show that the model returned by REWARDGUIDEDFINETUNINGSOLVER via SOC *necessarily encodes* score information.

**Theorem 6.1** (SOC Retains Score Information)**.** *Suppose the forward process in Equation (20) maps any distribution to standard Gaussian noise (i.e., a standard assumption in diffusion model literature). Then the solution to Equation (19) is $u^\star(x, t) := \sigma(t)\nabla \log p_t^k(x)$, where $p_t^k$ denotes the marginal distribution of the forward process in Equation (20), initialized at $p_1^{\pi_k}$. In other words,* REWARDGUIDEDFINETUNINGSOLVER *exactly recovers the score function.*

Leveraging the established connection between Eq. 19 and *mirror descent* (Tang, 2024), Theorem 6.1 enables us to reinterpret Algorithm 1 as generating *approximate mirror iterates*, a framework that has proven effective for sampling and generative modeling (Karimi et al., 2024; De Santi et al., 2025a;b).

**Robust Convergence under Inexact Updates.** Thanks to Theorem 6.1, we can now develop a rigorous convergence theory for Algorithm 1 under the realistic condition that REWARDGUIDEDFINETUNINGSOLVER (see Sec. 4) is implemented *approximately*. Let $\mathcal{G}$ be the objective in Eq. 9. Via $\pi^k$, the iterates generated by Algorithm 1 induce a sequence of stochastic processes, denoted by $\mathbf{Q}^k$, which satisfy $\mathbf{Q}^k = p_1^{\pi^k}$. Each iterate $\mathbf{Q}^k$ is understood as an approximation to the *idealized* mirror descent step:

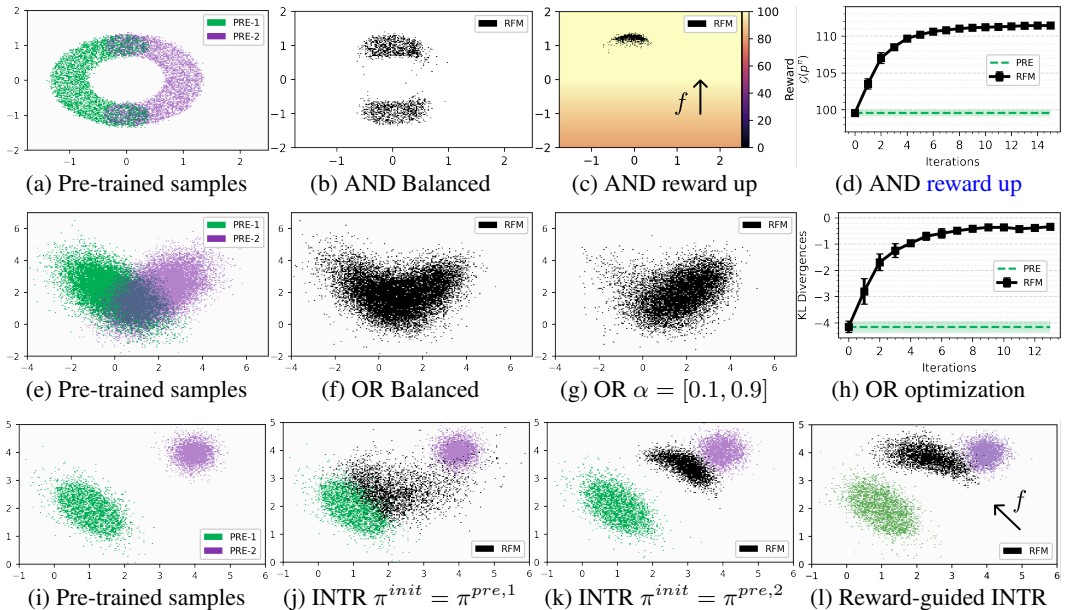

Figure 2: Illustrative settings with visually interpretable results. (top) Flow model balanced pure intersection (2b), and reward-guided intersection (2c), (mid) Flow balanced and unbalanced union, (bottom) Flow model pure and reward-guided interpolation. Crucially, RFM can correctly implement these practically relevant and diverse operators with high degree of expressivity (e.g., $\alpha$, reward-guidance).

$$\mathbf{Q}_\sharp^k \in \underset{\mathbf{Q}:p_0=p_0^{pre}}{\arg\max} \left\{ \langle \delta\mathcal{G}(p_1^{\pi_k}), \mathbf{Q} \rangle - \frac{1}{\gamma^k} D_{KL}\big(\mathbf{Q} \,\|\, \mathbf{Q}^{k-1}\big) \right\}. \tag{21}$$

which serves as the exact reference point for our analysis. To quantify the discrepancy between $\mathbf{Q}^k$ and $\mathbf{Q}_\sharp^k$, let $\mathcal{T}_k$ denote the history up to step $k$, and decompose the error as

$$b_k := \mathbb{E}\big[\delta\mathcal{G}(p_1^{\pi_k}) - \delta\mathcal{G}((\mathbf{Q}_\sharp^k)_1) \,\big|\, \mathcal{T}_k\big], \tag{22}$$

$$U_k := \delta\mathcal{G}(p_1^{\pi_k}) - \delta\mathcal{G}((\mathbf{Q}_\sharp^k)_1) - b_k. \tag{23}$$

Here, $b_k$ captures systematic approximation error, while $U_k$ represents a zero-mean fluctuation conditional on $\mathcal{T}_k$. Under mild assumptions controlling noise and bias (see Appendix B.2), the long-term behavior of the iterates can be rigorously characterized.

**Theorem 6.2** (Asymptotic convergence under inexact updates (Informal)). *Assume the oracle has bounded variance and diminishing bias, and the step sizes $\{\gamma^k\}$ satisfy the Robbins–Monro conditions ($\sum_k \gamma^k = \infty$, $\sum_k (\gamma^k)^2 < \infty$). Then the sequence $\{p_1^{\pi_k}\}$ generated by Algorithm 1 converges almost surely to the optimum in the weak sense:*

$$p_1^{\pi_k} \rightharpoonup p_1^* \quad a.s., \tag{24}$$

*where $p_1^* = \mathbf{Q}_1^*$, $\mathbf{Q}^* \in \arg\max_{\mathbf{Q}:\mathbf{Q}_0=p_0^{pre}} \mathcal{G}(\mathbf{Q}_1)$.*

## 7 EXPERIMENTAL EVALUATION

We evaluate RFM for the reward-guided flow merging problem (see Eq. 5) by tackling two types of experiments: (i) illustrative settings with visually interpretable insights, showcasing the correctness and high expressivity of RFM, and (2) high-dimensional molecular design tasks generating low-energy molecular conformers. Additional experimental details are reported in Appendix G.2

**Intersection Operator $\mathcal{O}_\wedge$ (AND).** We consider pre-trained flow models inducing densities $p_1^{pre,1}$ (green) and $p_1^{pre,2}$ (violet) - as shown in Fig. 2a. We fine-tune $\pi^{init} := \pi^{pre,1}$ via RFM to compute the policy $\pi^*$ resulting from diverse intersection operations $\pi^* = \mathcal{O}_\wedge(\pi^{pre,1}, \pi^{pre,2})$. First, in Fig. 2b, we show $p^*$ (black) obtained by RFM with $\alpha = [0.5, 0.5]$, i.e., *balanced* (B). One can notice that the flow model $p^*$ covers mostly the intersecting regions between $p_1^{pre,1}$ and $p_1^{pre,2}$ (see Fig. 2a).

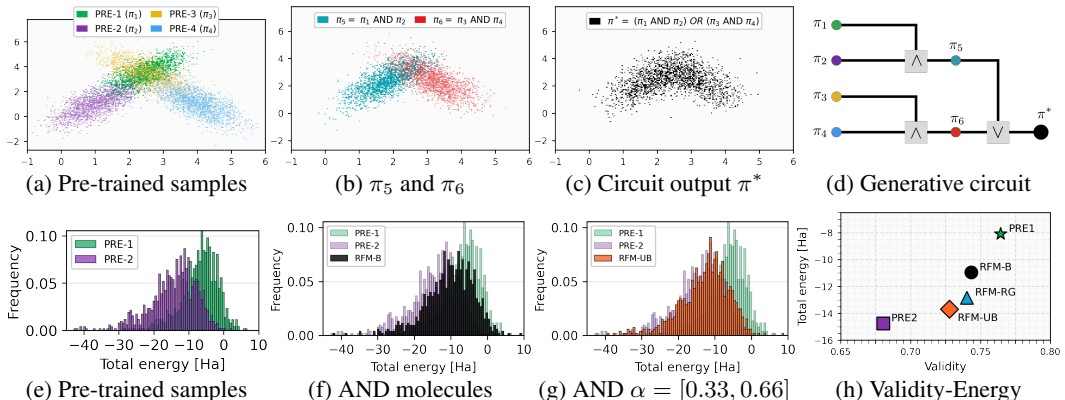

(a) Pre-trained samples     (b) $\pi_5$ and $\pi_6$     (c) Circuit output $\pi^*$     (d) Generative circuit

(e) Pre-trained samples     (f) AND molecules     (g) AND $\alpha = [0.33, 0.66]$     (h) Validity-Energy

Figure 3: (top) RFM can implement generative circuits (3d) computing sequential operators (3a-3c). (middle) RFM computes a flows intersection $\pi^*$ generating drug molecules with desired energy levels.

In Fig. 2c we report an instance of reward-guided intersection (RG) for a reward function maximized upward. As one can see, RFM computes a policy $\pi^*$ placing density over the highest-reward region among the intersecting ones, i.e., the top intersecting area. This reward-guided flow merging process is carried out via maximization over $K = 15$ iterations of the objective $\mathcal{G}$ illustrated in Fig. 2d.

**Union Operator $\mathcal{O}_\vee$ (OR).** We fine-tune the pre-trained flow model $\pi^{init} = \pi^{pre,1}$ with density illustrated in Fig. 2e (green) via RFM to implement balanced (i.e., $\alpha = [0.5, 0.5]$ and unbalanced (i.e., $\alpha = [0.1, 0.9]$ (UB)) versions of the union operator, namely computing $\pi^* = \mathcal{O}_\vee(\pi^{pre,1}, \pi^{pre,2})$. As shown in Fig. 2f and 2g RFM can successfully compute optimal policies $\pi^*$ implementing both operators via optimization of the functional $\mathcal{G}$, corresponding to sum of weighted KL-divergences (see Eq. 7) evaluated for iterations $k \in [K]$ with $K = 13$ in Fig. 2h.

**Interpolation Operator $\mathcal{O}_{W_1}$ (Wasserstein-1 Barycenter).** We use RFM to compute flow models $\pi^*$ inducing densities $p_1^*$ corresponding to diverse interpolations between the the pre-trained models' densities illustrated in Fig. 2i. Although the optimal policy to which RFM converges asymptotically is invariant w.r.t. the initial flow model $\pi^{init}$ chosen for fine-tuning, here we show that this choice can actually be used to control the algorithm execution over few iterations (i.e., $K = 6$). As one can expect, Fig. 2j and 2k show that the result density after $K = 6$ iterations is closer to the flow model chosen as $\pi^{init}$, namely $\pi^{pre,1}$ (green) in Fig. 2j and $\pi^{pre,2}$ (violet) in Fig. 2k. We illustrate in Fig. 2l the density (black) obtained via reward-guided interpolation, with a reward function maximized left upwards.

**Complex Logic Expressions via Generative Circuits.** We consider 4 flow models $\{\pi_{pre,i}\}_{i=1}^4$ illustrated in Fig. 3a, which we aim to merge into a unique flow $\pi^*$ determined by the logical expression $\pi^* = (\pi_1 \wedge \pi_2) \vee (\pi_3 \wedge \pi_4)$. In particular, we implement the generative circuit shown in Fig. 3d via sequential use of RFM. First, we compute $\pi_5 := \mathcal{O}_\wedge(\pi^{pre,1}, \pi^{pre,2})$ and $\pi_6 := \mathcal{O}_\wedge(\pi^{pre,3}, \pi^{pre,4})$, shown in Fig. 3b, and subsequently $\pi^* := \mathcal{O}_\vee(\pi^{pre,3}, \pi^{pre,4})$ - this is illustrated in Fig. 3c. Crucially, this illustrative experiments confirms that RFM can implement complex logical expressions over generative models via generative circuits, as the simple one just presented.

**Low-Energy Molecular Design via Flow Merging** Next, we address a de-novo molecular design task. Efficiently navigating the vast chemical space to discover novel structures with targeted physicochemical properties is a central goal of data-driven molecular design. A generative model must therefore be capable of producing diverse, chemically valid structures that follow specified property profiles and constraints. We base our case study on two FlowMol models $\pi^{pre,1}$ and $\pi^{pre,2}$ (Dunn & Koes, 2024) pre-trained on GEOM-Drugs (Axelrod & Gomez-Bombarelli, 2022) with different levels of single-point total energy at the GFN1-xTB level of theory (Friede et al., 2024), $-14.8$ and $-8.1$ Ha respectively as shown in Fig. 3e. We aim to compute a flow model that generates molecules whose total energy matches that of molecules likely under both generative models. To this end, we run RFM to compute the flow $\pi^*$ returned by the intersection operator (see Eq. 6), with parameters detailed in Apx. G.2. We report in Fig. 3f the

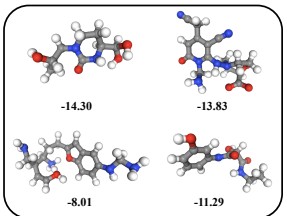

Figure 4: Drug molecules generated by $\pi_{AND}^*$ flow.

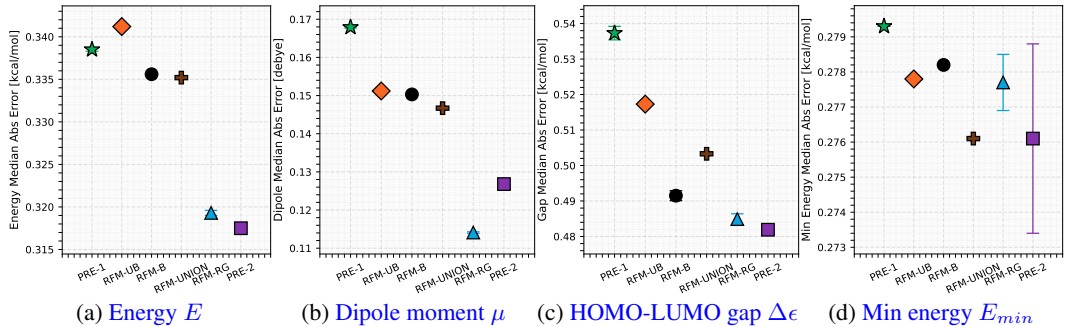

(a) Energy $E$  (b) Dipole moment $\mu$  (c) HOMO-LUMO gap $\Delta\epsilon$  (d) Min energy $E_{min}$

Figure 5: RFM can perform balanced (B), unbalanced (UB), reward-guided (RG) intersections, as well as unions (UNION) of prior ETFlow (Hassan et al., 2024) conformer generation models. We evaluate the resulting flow models in terms of energy (5a), dipole moment (5b), HOMO–LUMO gap (5c), and minimum energy (5d). These results demonstrate the ability of RFM to compute new flow models whose properties predictably interpolate those of the available pre-trained flows.

density $p^*$ (black) computed via balanced merging (i.e., $\alpha_1 = \alpha_2 = 1$) and in Fig. 3g the one obtained via unbalanced merging (i.e., $\alpha_1 = 1, \alpha_2 = 2$). In the former case, $p^*$ correctly places the majority of its density on energy levels within $[-20, 0]$ Ha (see Fig. 3f) corresponding to the overlapping region between the two priors. Moreover, the estimated mean energy of $\pi^*$ (black) i.e., $-10.95 \pm 0.28$ Ha, reported along with validity in 3h, nearly-perfectly matches the energy value of maximal overlap between $\pi^{pre,1}$ and $\pi^{pre,2}$, as one can see in 3e. Furthermore, adding reward-guidance leads to lower energy values in comparison to the balanced merging model while keeping its high validity. We show in Fig. 4 a sample of molecules generated via $\pi^*$, along with their total energy. In the unbalanced case, RFM shifts the density slightly leftwards, effectively implementing the $\alpha$-weighted intersection. We report energy-validity metrics resulting from balanced and unbalanced intersection in Fig. 3h, and compare them with their reward-guided counterpart in Table 1. Next, we compute via RFM the union operator over two FlowMol pre-trained on the QM9 dataset (Ramakrishnan et al., 2014). We parametrize critics $\phi_i^*$ (see Sec. 1) via the FlowMol latent representation with an MLP readout layer. Figure 7 shows that the estimated mean of the model $\pi^*$ obtained via RFM matches the average total energy of $\pi^{pre,1}$ and $\pi^{pre,2}$ as predicted by the closed-form expression for union from Sec. 3.

**Flow Merging of Conformer Generation Models** Lastly, we tackle a conformer generation task. Deriving 3D conformers from the molecule's topology is a key prerequisite for many computational chemistry applications spanning molecular docking (McNutt et al., 2023), thermodynamic property prediction (Pracht & Grimme, 2021), and modeling reaction pathways for catalyst design (Schmid et al., 2025), among others. Given a molecular graph, a good conformer generator should predict 3D structures that $(i)$ cover the entire ensemble of chemically valid structures that can be observed in nature for that molecule and $(ii)$ generate those structures at their local energy minimum. In this work, we leverage the pre-trained GEOM-QM9 ETFlow model (denoted PRE-1) (Hassan et al., 2024). Due to ETFlows' already high coverage, we choose to evaluate our method on energetic ensemble properties, as presented in Hassan et al. (2024). Specifically, for a given molecule we generate a set of conformers and measure the difference in energy, dipole moment, HOMO-LUMO gap and minimum energy of the generated structure ensemble compared to the equilibrium ensemble.

We obtain a lower-energy model PRE-2 via AM fine-tuning on the negative GFN1-xTB total energy, like in our de-novo molecular design case study (see G.3). Afterwards, we use RFM initialized from PRE-2, to compute its balanced (B), unbalanced (UB), reward-guided (RG) intersection, and union variants. Figure 5a shows that the median absolute error (MAE) on the total energy $E$ smoothly interpolates between PRE-1 ($\approx 0.3385$ kcal/mol) and PRE-2 ($\approx 0.3175$ kcal/mol): RFM-B and RFM-UNION achieve intermediate errors of $\approx 0.3356$ and $0.3352$ kcal/mol as expected. On the other hand, the reward-guided variant reaches $\approx 0.3193$ kcal/mol, close to PRE-2, and the unbalanced variant ($\alpha_1 = 0.7, \alpha_2 = 0.3$) remains near PRE-1 at $0.3412$ kcal/mol. These numerical results further validate the ability of RFM to perform unbalanced (UB) and reward-guided (RG) intersection, leading to flows with properties controllably interpolating the ones of available flow models. A similar pattern appears for the dipole moment $\mu$ in Fig. 5b where PRE-1 and PRE-2 attain MAEs of $\approx 0.1679$ and $0.1268$ debye respectively. The merged models lie between these values, with

RFM-RG further reducing the error to $\approx 0.1141$ debye. Analogous results are reported in Fig. 5c and 5d for the HOMO–LUMO gap $\Delta\epsilon$, and minimum energy $E_{\min}$. This evaluation indicates that RFM can compute new flow models for conformer generation, whose physical properties controllably and predictably interpolate between, or slightly improve upon, the two available pre-trained flow models.

Ultimately, in Apx. F, we briefly investigate the computational cost of Reward-Guided Flow Merging.

## 8 RELATED WORK

In the following, we present relevant work in related areas, including flow model fine-tuning via optimal control, flow model merging and composition, convex RL, and probability-space optimization.

**Flow and diffusion models fine-tuning via optimal control.** Several works have framed fine-tuning of flow and diffusion models to maximize expected reward functions under KL regularization as an entropy-regularized optimal control problem (e.g., Uehara et al., 2024a; Tang, 2024; Uehara et al., 2024b; Domingo-Enrich et al., 2024). More recently, De Santi et al. (2025b) introduced a framework for distributional fine-tuning. The reward-guided flow merging problem in Eq. 5 extends a specific sub-class of distributional fine-tuning to the case of multiple (i.e., $n > 1$) pre-trained models. This generalization allows the use of scalable control theoretic or RL schemes for flow model merging, and enables reward-guided model merging, where reward-guided fine-tuning and model merging can be performed simultaneously via unified formulations and algorithms, such as RFM.

**Diffusion and flow model merging and inference-time composition.** While recent works in inference-time flow and diffusion model composition introduced theory-backed schemes (e.g., Skreta et al., 2024; Bradley et al., 2025; Du et al., 2023), this is arguably not the case for flow merging, with a few exceptions (e.g., Song et al., 2023). Our framework provides a formal probability-space viewpoint enabling interpretable merging operators (see Sec. 3) for highly expressive compositions (e.g., via generative circuits), provably implemented by RFM. To our knowledge, the theoretical guarantees in Sec. 6 are first-of-their-kind for model merging. Specializing them to specific operators e.g., intersection, yields highly relevant insights, such as generative models safety guarantees via intersection with a prior safe model.

**Convex and general utilities reinforcement learning.** Convex and General (Utilities) RL (Hazan et al., 2019; Zahavy et al., 2021; Zhang et al., 2020) generalizes RL to the case where one wishes to maximize a concave (Hazan et al., 2019; Zahavy et al., 2021), or general (Zhang et al., 2020; Barakat et al., 2023) functional of the state distribution induced by a policy over a dynamical system's state space. Recent works tackled the finite samples budget setting (e.g., Mutti et al., 2022b;a; De Santi et al., 2024). Similarly to previous optimization schemes for diffusion and flow models (De Santi et al., 2025a;b), our framework (in Eq. 5) is related to Convex and General RL, with $p_1^\pi$ representing the state distribution induced by policy $\pi$ over a subset, or the entire flow process state space.

**Optimization over probability measures via mirror flows.** Recently, there has been a growing interest in devising theoretical guarantees for probability-space optimization problems in diverse fields of application. These include optimal transport (Aubin-Frankowski et al., 2022; Léger, 2021; Karimi et al., 2024), kernelized methods (Dvurechensky & Zhu, 2024), GANs (Hsieh et al., 2019), and manifold exploration (De Santi et al., 2025a) among others. To our knowledge, we present the first use of this theoretical framework to establish guarantees for large-scale flow and diffusion models merging, shedding new light on this highly practically relevant generative modeling task.

## 9 CONCLUSION

This work introduces a formal probability-space optimization framework for reward-guided flow merging, strictly generalizing existing formulations. This allows to express a rich class of practically relevant merging operators over generative models (e.g., intersection, union, interpolation, as well as their reward-guided counterparts), as well as complex logical expressions via generative circuits. We then propose Reward-Guided Flow Merging, a mirror-descent algorithm that reduces complex merging tasks to a sequence of standard reward-guided fine-tuning steps, each solvable by scalable off-the-shelf methods. Leveraging recent advances in mirror flows theory, we provide first-of-their kind guarantees for (reward-guided) flow model merging. Empirical results on diverse visually interpretable settings, molecular design as well as conformer generation tasks demonstrate that our approach can steer pre-trained models to implement diverse reward-guided merging objectives of high practical relevance.

## 10 REPRODUCIBILITY STATEMENT

We provide details explanation of the method proposed in Sec. 4 and conditions under which it work in Sec. 3. We include in Appendix E.2 a detailed implementation, which we used to carry our the experiments in Sec. 7. Moreover, we report parameter choices for experimental evaluations in Apx. G.2. Ultimately, notice that our implemented version of RFM is based on Adjoint Matching (Domingo-Enrich et al., 2024), which is a established scheme for reward-guided fine-tuning.

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

# A APPENDIX

## CONTENTS

## B    PROOFS FOR SECTION 6

### B.1    PROOF OF THEOREM 6.1

**Stochastic Optimal Control.**    We consider stochastic optimal control (SOC), which studies the problem of steering a stochastic dynamical system to optimize a specified performance criterion. Formally, let $(X_t^u)_{t \in [0,1]}$ be a controlled stochastic process satisfying the stochastic differential equation (SDE)

$$\mathrm{d}X_t^u = b(X_t^u, t)\, \mathrm{d}t + \sigma(t)\, u(X_t^u, t)\, \mathrm{d}t + \sigma(t)\, \mathrm{d}B_t, \qquad X_0^u \sim p_0,$$

where $u \in \mathcal{U}$ is an admissible control and $B_t$ is standard Brownian motion. The objective is to select $u$ to minimize the cost functional

$$\mathbb{E}\left[ \int_0^1 \frac{1}{2}\|u(X_t^u, t)\|^2 \, dt - g(X_1^u) \right], \tag{25}$$

where $\frac{1}{2}\|u(\cdot, t)\|^2$ represents the running cost and $g$ is a terminal reward. A standard application of Girsanov's theorem shows that Equation (25) is equivalent to the mirror descent iterate in Equation (21) with $\delta \mathcal{G}(p_1^{\pi_k}) \leftarrow g$ and $p_0 \leftarrow p^{pre}$ (Tang, 2024). In addition, it is well-known that in the context of diffusion-based generative modeling, the corresponding uncontrolled dynamics

$$\mathrm{d}X_t = -b(X_t, t)\, \mathrm{d}t + \sigma(t)\, \mathrm{d}B_t$$

coincide with the forward noising process used in score-based models (Song et al., 2020; Domingo-Enrich et al., 2024).

**Proof of Theorem 6.1.**

**Theorem 6.1** (SOC Retains Score Information). *Suppose the forward process in Equation (20) maps any distribution to standard Gaussian noise (i.e., a standard assumption in diffusion model literature). Then the solution to Equation (19) is $u^\star(x, t) := \sigma(t)\, \nabla \log p_t^k(x)$, where $p_t^k$ denotes the marginal distribution of the forward process in Equation (20), initialized at $p_1^{\pi_k}$. In other words,* REWARDGUIDEDFINETUNINGSOLVER *exactly recovers the score function.*

*Proof.*    **Step 1.** Let $\mathbf{Q}^\star$ denote the optimal process solving Equation (19). A standard application of Girsanov's theorem shows that $\mathbf{Q}^\star$ also solves the *Schrödinger bridge problem*

$$\min_{\substack{\mathbf{Q}_0 = p^{\mathrm{pre}} \\ \mathbf{Q}_1 = \mathbf{Q}_1^\star}} D_{\mathrm{KL}}\big(\mathbf{Q} \,\|\, \mathbf{P}\big), \tag{26}$$

where $\mathbf{P}$ is the law of the uncontrolled dynamics

$$\mathrm{d}X_t = b(X_t, t)\, \mathrm{d}t + \sigma(t)\, \mathrm{d}B_t.$$

This equivalence holds because the SOC cost in Equation (19) penalizes control energy in the same way that Girsanov's theorem expresses a controlled SDE as a relative entropy with respect to its uncontrolled counterpart.

**Step 2.** Define the *forward process* $\mathbf{P}_{\mathrm{forward}}$ by

$$\mathrm{d}X_t = -b(X_t, t)\, \mathrm{d}t + \sigma(t)\, \mathrm{d}B_t. \tag{27}$$

By assumption, this process maps any initial distribution to the standard Gaussian at $t = 1$. In particular, starting from $X_0 \sim \mathbf{Q}_1^\star$, we obtain $X_1 \sim p^{\mathrm{pre}} = \mathcal{N}(0, I)$.

**Step 3.** Consider the *time-reversed Schrödinger bridge problem*

$$\min_{\substack{\overleftarrow{\mathbf{Q}}_0 = \mathbf{Q}_1^\star \\ \overleftarrow{\mathbf{Q}}_1 = p^{\mathrm{pre}}}} D_{\mathrm{KL}}\big(\overleftarrow{\mathbf{Q}} \,\|\, \mathbf{P}_{\mathrm{forward}}\big), \tag{28}$$

and denote its solution by $\overleftarrow{\mathbf{Q}}^\star$. Since relative entropy is invariant under bijective mappings and time-reversal is bijective, the optimizers of Equation (26) and Equation (28) satisfy

$$\overleftarrow{\mathbf{Q}^\star} = \overleftarrow{\mathbf{Q}^\star}$$

i.e., the optimal reversed bridge is simply the time-reversal of the forward bridge.

By **Step 2**, the process

$$\mathrm{d}X_t = -b(X_t, t)\,\mathrm{d}t + \sigma(t)\,\mathrm{d}B_t, \qquad X_0 \sim \mathbf{Q}_1^\star \tag{29}$$

solves Equation (28), achieving the minimum relative entropy (zero) while satisfying the prescribed marginals. Thus, invoking the relation $\overleftarrow{\mathbf{Q}}^\star = \overleftarrow{\mathbf{Q}}^\star$, the solution to Equation (26)—and hence to the SOC problem Equation (19)—is given by the time-reversal of Equation (29).

Finally, applying the classical time-reversal formula (Anderson, 1982) yields that $\mathbf{Q}^\star$ is given by

$$\mathrm{d}X_t = \left( b(\overleftarrow{X}_t, t) + \sigma^2(t)\,\nabla \log p_t(X_t) \right) \mathrm{d}t + \sigma(t)\,\mathrm{d}B_t,$$

where $p_t$ is the marginal density of Equation (29). Hence, REWARDGUIDEDFINETUNINGSOLVER exactly recovers the score function. □

## B.2 RIGOROUS STATEMENT AND PROOF OF THEOREM 6.2

To prepare for the convergence analysis, we impose a few auxiliary assumptions. These assumptions are standard in the study of stochastic approximation and gradient flows, and typically hold in practical situations. Our proof strategy follows ideas that have also been employed in related works (De Santi et al., 2025a;b).

We begin with the entropy functional defined on probability measures:

$$\mathcal{H}(p) := \int p \log p. \tag{30}$$

In our analysis, $\mathcal{H}$ serves as the *mirror map* or *distance-generating function* (Mertikopoulos et al., 2024; Hsieh et al., 2019). The first condition addresses the behavior of the corresponding dual variables.

**Assumption B.1** (Precompactness of Dual Iterates). *The sequence of dual elements $\{\delta\mathcal{H}(p_1^{\pi_k})\}_k$ is precompact in the $L_\infty$ topology.*

This compactness property ensures that the interpolated dual trajectories remain confined to a bounded region of function space. Such a condition is crucial for invoking convergence results based on asymptotic pseudotrajectories. Variants of this assumption have appeared in the literature on stochastic approximation and continuous-time embeddings of discrete algorithms (Benaïm, 2006; Hsieh et al., 2019; Mertikopoulos et al., 2024).

**Assumption B.2** (Noise and Bias Conditions). *For the stochastic approximations used in the updates, we assume that almost surely:*

$$\|b_k\|_\infty \to 0, \tag{31}$$

$$\sum_k \mathbb{E}\left[ \gamma_k^2 \left( \|b_k\|_\infty^2 + \|U_k\|_\infty^2 \right) \right] < \infty, \tag{32}$$

$$\sum_k \gamma_k \|b_k\|_\infty < \infty. \tag{33}$$

These conditions, standard in the Robbins–Monro setting (Robbins & Monro, 1951; Benaïm, 2006; Hsieh et al., 2019), guarantee that the stochastic bias vanishes asymptotically while the cumulative noise remains under control. Together, they ensure that random perturbations do not obstruct convergence to the optimizer of the limiting objective.

With these assumptions in place, we can now state and prove the convergence guarantee.

**Theorem B.1** (Convergence guarantee in the trajectory setting). *Suppose Assumptions B.1–B.2 hold, and the step sizes $\{\gamma_k\}$ follow the Robbins–Monro conditions ($\sum_k \gamma_k = \infty$, $\sum_k \gamma_k^2 < \infty$). Then the sequence $\{p_1^{\pi_k}\}$ generated by Algorithm 1 converges almost surely, in the weak topology, to the optimum:*

$$p_1^{\pi_k} \rightharpoonup p_1^* \quad a.s., \tag{34}$$

*where $p_1^* = \mathbf{Q}_1^*$ for some $\mathbf{Q}^* \in \arg\max_{\mathbf{Q}:\mathbf{Q}_0 = p_0^{pre}} \mathcal{G}(\mathbf{Q}_1)$.*

*Proof.* We analyze the continuous-time mirror flow defined by

$$\dot{h}_t = \delta\mathcal{G}(p_1^t), \qquad p_1^t = \delta\mathcal{H}^\star(h_t), \tag{35}$$

where the Fenchel conjugate of $\mathcal{H}$ is given by $\mathcal{H}^\star(h) = \log \int e^h$ (Hsieh et al., 2019; Hiriart-Urruty & Lemaréchal, 2004).

To link the discrete dynamics to this continuous flow, we construct a piecewise linear interpolation of the iterates:

$$\hat{h}_t = h^{(k)} + \frac{t - \tau_k}{\tau_{k+1} - \tau_k}\big(h^{(k+1)} - h^{(k)}\big), \quad h^{(k)} = \delta\mathcal{H}(p_1^{\pi_k}), \quad \tau_k = \sum_{r=0}^{k} \alpha_r,$$

where $\{\alpha_r\}$ denotes the step-size sequence. This interpolation produces a continuous path $\hat{h}_t$ that tracks the discrete updates as the steps shrink.

Let $\Phi_u$ denote the flow map of equation 35 at time $u$. Standard results in stochastic approximation (Benaïm, 2006; Hsieh et al., 2019; Mertikopoulos et al., 2024) imply that for any fixed horizon $T > 0$, there exists a constant $C(T)$ such that

$$\sup_{0 \le u \le T} \|\hat{h}_{t+u} - \Phi_u(\hat{h}_t)\| \le C(T)\Big[\Delta(t-1, T+1) + b(T) + \gamma(T)\Big],$$

where $\Delta$ accounts for cumulative noise, $b$ for bias, and $\gamma$ for step-size effects. Under Assumptions B.1–B.2, these quantities vanish asymptotically, ensuring that $\hat{h}_t$ forms a precompact asymptotic pseudotrajectory (APT) of the mirror flow.

By the APT limit set theorem (Benaïm, 2006, Thm. 4.2), the limit set of a precompact APT is contained in the internally chain transitive (ICT) set of the underlying flow. In our case, Equation (35) corresponds to a gradient-like flow in the Hellinger–Kantorovich geometry (Mielke & Zhu, 2025), with $\mathcal{G}$ serving as a strict Lyapunov function. As $\mathcal{G}$ decreases strictly along non-stationary trajectories, the ICT set reduces to the collection of stationary points of $\mathcal{G}$.

Finally, because $\mathcal{G}$ is composed of distance-like penalties (e.g., $\mathbb{W}_1$ or KL terms) together with a linear component, its stationary points coincide with its global maximizers. Consequently, $\hat{h}_t$ converges almost surely to the set of maximizers of $\mathcal{G}$, which establishes the claim. $\square$

## C  DERIVATIONS OF GRADIENTS OF FIRST VARIATION

### C.1  A BRIEF TUTORIAL ON FIRST VARIATION DERIVATION

In this work, we focus on the functionals that are Fréchet differentiable: Let $V$ be a normed spaces. Consider a functional $F : V \to \mathbb{R}$. There exists a linear operator $A : V \to \mathbb{R}$ such that the following limit holds

$$\lim_{\|h\|_V \to 0} \frac{|F(f + h) - F(f) - A[h]|}{\|h\|_V} = 0. \tag{36}$$

We further assume that $V$ has enough structure such that every element of its dual (the space of bounded linear operator on $V$) admits a compact representation. For example, if $V$ is the space of bounded continuous functions with compact support, there exists a unique positive Borel measure $\mu$ with the same support, which can be identified as the linear functional. We denote this element as $\delta F[f]$ such that $\langle \delta F[f], h \rangle = A[h]$. Sometimes we also denote it as $\frac{\delta F}{\delta f}$. We will refer to $\delta F[f]$ as the first-order variation of $F$ at $f$.

In the following, we briefly present standard strategies to derive the first-order variation of two broad classes of functionals, including a wide variety of divergence measures, which can be employ to implement novel operators by Eq. 5. We consider: $(i)$ those defined in closed form with respect to the density (e.g., forward KL) and, $(ii)$ those defined via variational formulations (e.g., Wasserstein distance, reverse KL, and MMD).

- **Category 1: Functional defined in a closed form with respect to the density.** For this class of functionals, the first-order variations can typically be computed using its definition and chain rule. Recalling the definition of first variation (36), we can calculate the first-order variation of the mean functional, as a trivial example. Given a continuous and bounded function $r : \mathbb{R}^d \to \mathbb{R}$ and a probability measure $\mu$ on $\mathbb{R}^d$, define the functional $F(\mu) = \int r(x)\mu(x)dx$. Then we have:

$$|F(\mu + \delta\mu) - F(\mu) - \langle r, \delta\mu \rangle| = 0. \tag{37}$$

Therefore we obtain that: $\delta F[\mu] = r$ for all $\mu$. In the following section, we compute similarly the first variation of the KL divergence.

- **Category 2: Functionals defined through a variational formulation.** Another fundamental subclass of functionals that plays a central role in this work is the one of functionals defined via a variational problem

$$F[f] = \sup_{g \in \Omega} G[f, g], \tag{38}$$

where $\Omega$ is a set of functions or vectors independent of the choice of $f$, and $g$ is optimized over the set $\Omega$. We will assume that the maximizer $g^*(f)$ that reaches the optimal value for $G[f, \cdot]$ is unique (which is the case for the functionals considered in this project). It is known that one can use the Danskin's theorem (also known as the envelope theorem) to compute

$$\frac{\delta F[f]}{\delta f} = \partial_f G[f, g^*(f)], \tag{39}$$

under the assumption that $F$ is differentiable (Milgrom & Segal, 2002).

### C.2  DERIVATION OF FIRST VARIATIONS USED IN SEC. 4

In the following, we derive explicitly the first variations employed in Sec. 1

- **Optimal transport and Wasserstein-p distance (Category 2)** Consider the optimal transport problem

$$\mathrm{OT}_c(u, v) = \inf_\gamma \left\{ \int \int \int c(x, y)d\gamma(x, y) : \int \gamma(x, y)dx = u(y), \int \gamma(x, y)dy = v(x) \right\} \tag{40}$$

where

$$\Gamma = \left\{ \gamma : \int \gamma(x, y)dx = u(y), \int \gamma(x, y)dy = v(x) \right\}$$

It admits the following equivalent dual formulation

$$\text{OT}_c(u, v) = \sup_{f,g} \left\{ \int f du + \int g dv : f(x) + g(y) \leq c(x, y) \right\} \tag{41}$$

By taking $c(x, y) = \|x - y\|^p$, we recover $\text{OT}_c(u, v) = W_p(u, v)^p$. Let $\phi^*$ and $g^*$ be the solution to the above dual optimization problem. From the Danskin's theorem, we have

$$\frac{\delta}{\delta u} W_p(u, v)^p = \phi^*. \tag{42}$$

In the special case of $p = 1$, we know that $g^* = -\phi^*$ (note that the constraint can be equivalently written as $\|\nabla \phi\| \leq 1$), in which case $\phi^*$ is typically known as the critic in the Wasserstein-GAN framework (cf. Arjovsky et al., 2017).

- **Reverse KL divergence (Category 2)** We use the variational (Fenchel–Legendre) representation of the forward KL, $D_{KL}(p\|q)$, as in f-GAN (Nowozin et al., 2016):

$$D_{KL}(p\|q) = \sup_{\phi:\mathcal{X}\to\mathbb{R}} \left\{ \mathbb{E}_p \phi(x) - \mathbb{E}_q e^{\phi(x)-1} \right\} \tag{43}$$

which follows from the general f-divergence dual generator $f(u) = u \log u - u + 1$ whose conjugate is $f^*(t) = e^{t-1}$. For fixed $p$ and variable $q$, we define:

$$G(q, \phi) := \mathbb{E}_p \phi(x) - \mathbb{E}_q e^{\phi(x)-1} \tag{44}$$

Assuming uniqueness of a maximizer $\phi^*(p, q)$, Danskin's (or envelope) theorem yields the first variation by differentiating $G$ at $\phi^*$:

$$\frac{\delta}{\delta q(x)} D_{KL}(p\|q) = \frac{\delta}{\delta q(x)} \left( -\int q(x) e^{\phi^*(x)-1} du \right) = -e^{\phi^*(x)-1} \tag{45}$$

- **KL divergence (Category 1)** Consider the KL functional:

$$D_{KL}(p\|q) = -\int p \log \frac{p}{q}, dx \tag{46}$$

By the definition of the first-order variation (see Eq. 36), we have:

$$\delta D_{KL}(p\|q) = \log \frac{p}{q} + 1 \tag{47}$$

# D    PROOF OF PROPOSITION 1

**Proposition 1** (Union operator via Pre-trained Mixture Density Representation). *Given $\bar{p}_1^{pre} = \sum_{i=1}^{n} \alpha_i p_1^{pre,i} / \sum_{i=1}^{n} \alpha_i$, i.e., the $\alpha$-weighted mixture density of pre-trained models, the following hold:*

$$\pi^* \in \arg\min_\pi \sum_{i=1}^{n} \alpha_i \, D_{KL}^R(p_1^\pi \,\|\, p_1^{pre,i}) = \left( \sum_{i=1}^{n} \alpha_i \right) D_{KL}^R(p_1^\pi \,\|\, \bar{p}_1^{pre}) \tag{14}$$

*Proof.* We prove the statement for $n = 2$, which trivially generalizes to any $n$. We first rewrite the LHS optimization problem as:

$$\arg\min_\pi \mathcal{F}(p^\pi) \tag{48}$$

where we denote $p_1^\pi$ by $p^\pi$ for notational concision and define $p_1 = p^{pre,i}$ and $p_2 = p^{pre,2}$. Then we have:

$$\mathcal{F}(p^\pi) = \alpha_1 \, \mathbb{E}_{p_1}[\log p_1 - \log p^\pi] + \alpha_2 \, \mathbb{E}_{p_2}[\log p_2 - \log p^\pi] \tag{49}$$

$$= \alpha_1 \, \mathbb{E}_{p_1} \log p_1 + \alpha_2 \, \mathbb{E}_{p_2} \log p_2 - \left( \alpha_1 \, \mathbb{E}_{p_1} \log p^\pi + \alpha_2 \, \mathbb{E}_{p_2} \log^\pi \right) \tag{50}$$

We now write the following, where $\bar{p}$ denotes $\bar{p}_1^{pre}$:

$$\mathbb{E}_{\bar{p}} \log p^\pi = \int \log p^\pi(x) \bar{p}(x) \, \mathrm{d}x \tag{51}$$

$$= \int \log p^\pi(x) \left[ \frac{\alpha_1 p_1}{\alpha_1 + \alpha_2} + \frac{\alpha_2 p_2}{\alpha_1 + \alpha_2} \right](x) \, \mathrm{d}x \tag{52}$$

$$= \frac{1}{\alpha_1 + \alpha_2} \left( \log p^\pi(x) \alpha_1 p_1(x) + \log p^\pi(x) \alpha_2 p_2(x) \right) \tag{53}$$

$$= \frac{1}{\alpha_1 + \alpha_2} \left( \alpha_1 \, \mathbb{E}_{p_1} \log p^\pi + \alpha_2 \, \mathbb{E}_{p_2} \log p^\pi \right) \tag{54}$$

By combining Eq. 50 and 54, we obtain:

$$\mathcal{F}(p^\pi) = \alpha_1 \, \mathbb{E}_{p_1} \log p_1 + \alpha_2 \, \mathbb{E}_{p2} \log p_2 - (\alpha_1 + \alpha_2) \, \mathbb{E}_{\bar{p}} \log p^\pi \tag{55}$$

Therefore,

$$\arg\min_\pi \mathcal{F}(p^\pi) = \arg\min_\pi \underbrace{\alpha_1 \, \mathbb{E}_{p_1} \log p_1 + \alpha_2 \, \mathbb{E}_{p_2} \log p_2}_{\text{constant}} - (\alpha_1 + \alpha_2) \, \mathbb{E}_{\bar{p}} \log p^\pi \tag{56}$$

$$= \arg\min_\pi -(\alpha_1 + \alpha_2) \, \mathbb{E}_{\bar{p}} \log p^\pi \tag{57}$$

$$= \arg\min_\pi -(\alpha_1 + \alpha_2) \, \mathbb{E}_{\bar{p}} \log p^\pi + \underbrace{(\alpha_1 + \alpha_2) \, \mathbb{E}_{\bar{p}} \log \bar{p}}_{\text{constant}} \tag{58}$$

$$= \arg\min_\pi (\alpha_1 + \alpha_2) D_{KL}(\bar{p} \| p^\pi) \tag{59}$$

$$\tag{60}$$

Which concludes the proof. $\qquad\square$

# E    REWARD-GUIDED FLOW MERGING (RFM) IMPLEMENTATION

In the following, we provide an example of detailed implementations for REWARDGUIDEDFINETUN-
INGSOLVER employed in Sec. 4 by Reward-Guided Flow Merging, as well as REWARDGUIDEDFINE-
TUNINGSOLVERRUNNINGCOSTS, leveraged in Sec. 5 to scalably implement the AND operator. While
the oracle implementation we report for completeness for REWARDGUIDEDFINETUNINGSOLVER corre-
sponds to classic Adjoint Matching (AM) (Domingo-Enrich et al., 2024), the one for REWARDGUID-
EDFINETUNINGSOLVERRUNNINGCOSTS trivially extends AM base implementation to account for the
running cost terms introduced in Eq. 17.

## E.1    IMPLEMENTATION OF REWARDGUIDEDFINETUNINGSOLVER

Before detailing the implementations, we briefly fix notation. Both algorithms explicitly rely on
the interpolant schedules $\kappa_t$ and $\omega_t$ from equation 1. In the flow-model literature, these are more
commonly denoted $\alpha_t$ and $\beta_t$. We write $u^{\text{pre}}$ for the velocity field induced by the pre-trained policy
$\pi^{\text{pre}}$, and $u^{\text{fine}}$ for the velocity field induced by the fine-tuned policy. In essence, each algorithm first
draws trajectories and then uses them to approximate the solution of a surrogate ODE; its marginals
serve as regression targets for the control policy (Section 5 Domingo-Enrich et al., 2024).

---

**Algorithm 2** REWARDGUIDEDFINETUNINGSOLVER via AM

---

**Require:** Pre-trained FM velocity field $u^{\text{pre}}$, step size $h$, number of fine-tuning iterations $N$, gradient
    of reward $\nabla r$, fine-tuning strength $\eta_k$
1:  Initialize fine-tuned vector fields: $u^{\text{finetune}} = u^{\text{pre}}$ with parameters $\theta$.
2:  **for** $n \in \{0, \dots, N-1\}$ **do**
3:      Sample $m$ trajectories $\boldsymbol{X} = (X_t)_{t \in \{0, \dots, 1\}}$ with memoryless noise schedule:

$$\sigma(t) = \sqrt{2\kappa_t \left( \frac{\dot{\omega}_t}{\omega_t} \kappa_t - \dot{\kappa}_t \right)} \tag{61}$$

4:      i.e.,:

$$X_{t+h} = X_t + h\left(2u_\theta^{\text{finetune}}(X_t, t) - \frac{\dot{\omega}_t}{\omega_t} X_t\right) + \sqrt{h}\,\sigma(t)\,\varepsilon_t, \quad \varepsilon_t \sim \mathcal{N}(0, I), \quad X_0 \sim \mathcal{N}(0, I). \tag{51}$$

5:      For each trajectory, solve the *lean adjoint ODE* backwards in time from $t = 1$ to $0$, e.g.:

$$\tilde{a}_{t-h} = \tilde{a}_t + h\,\tilde{a}_t^\top \nabla_{X_t}\left(2v^{\text{base}}(X_t, t) - \frac{\dot{\omega}_t}{\omega_t} X_t\right), \quad \tilde{a}_1 = \eta_k \nabla r(X_1). \tag{52}$$

6:      Note that $X_t$ and $\tilde{a}_t$ should be computed without gradients, i.e.,

$$X_t = \texttt{stopgrad}(X_t) \tag{62}$$
$$\tilde{a}_t = \texttt{stopgrad}(\tilde{a}_t) \tag{63}$$

7:      For each trajectory, compute the following Adjoint Matching objective:

$$\mathcal{L}_{\text{Adj-Match}}(\theta) = \sum_{t \in \{0, \dots, 1-h\}} \left\| \tfrac{2}{\sigma(t)}\left(v_\theta^{\text{finetune}}(X_t, t) - u^{\text{base}}(X_t, t)\right) + \sigma(t)\,\tilde{a}_t \right\|^2. \tag{53}$$

8:      Compute the gradient $\nabla_\theta \mathcal{L}(\theta)$ and update $\theta$ using favorite gradient descent algorithm.
9:  **end for**
**Output:** Fine-tuned vector field $v^{\text{finetune}}$

---

## E.2    IMPLEMENTATION OF REWARDGUIDEDFINETUNINGSOLVERRUNNINGCOSTS

The following REWARDGUIDEDFINETUNINGSOLVERRUNNINGCOSTS is algorithmically identical to
REWARDGUIDEDFINETUNINGSOLVER, with the only difference that the lean adjoint computation now

integrates a running-cost term $f_t$, defined as follows (see Sec. 5):

$$f_t(x) := \delta \left( \sum_{i=1}^{n} \alpha_i \, D_{KL}(p_t^\pi \,\|\, p_t^{pre,i}) \right)(x,t), \quad t \in [0,1) \tag{64}$$

---

**Algorithm 3** REWARDGUIDEDFINETUNINGSOLVERRUNNINGCOSTS via AM with running costs

---

**Require:** Pre-trained FM velocity field $v^{\text{base}}$, step size $h$, number of fine-tuning iterations $N$, $f_t = \nabla \delta \mathcal{G}_t(p_t^{\pi^k})$, weight $\gamma_k$, weight schedule $\lambda$

1: Initialize fine-tuned vector fields: $v^{\text{finetune}} = v^{\text{base}}$ with parameters $\theta$.

2: **for** $n \in \{0, \dots, N-1\}$ **do**

3:      Sample $m$ trajectories $\boldsymbol{X} = (X_t)_{t \in \{0,\dots,1\}}$ with memoryless noise schedule:

$$\sigma(t) = \sqrt{2\kappa_t \left( \frac{\dot{\omega}_t}{\omega_t} \kappa_t - \dot{\kappa}_t \right)} \tag{65}$$

4:      i.e.,:

$$X_{t+h} = X_t + h \left( 2 v_\theta^{\text{finetune}}(X_t, t) - \frac{\dot{\omega}_t}{\omega_t} X_t \right) + \sqrt{h} \, \sigma(t) \, \varepsilon_t, \quad \varepsilon_t \sim \mathcal{N}(0, I), \quad X_0 \sim \mathcal{N}(0, I). \tag{40}$$

5:      For each trajectory, solve the *lean adjoint ODE* backwards in time from $t = 1$ to $0$, e.g.:

$$\tilde{a}_{t-h} = \tilde{a}_t + h \, \tilde{a}_t^\top \nabla_{X_t} \left( 2 v^{\text{base}}(X_t, t) - \frac{\dot{\omega}_t}{\omega_t} X_t \right) - h \gamma_k \lambda_t f_t(X_t) \tag{66}$$

$$\tilde{a}_1 = -\gamma_k \lambda_1 \nabla_{X_1} \delta \mathcal{G}_1(p_1^{\pi^k})(X_1). \tag{41}$$

6:      Note that $X_t$ and $\tilde{a}_t$ should be computed without gradients, i.e.,

$$X_t = \texttt{stopgrad}(X_t) \tag{67}$$

$$\tilde{a}_t = \texttt{stopgrad}(\tilde{a}_t) \tag{68}$$

7:      For each trajectory, compute the Adjoint Matching objective:

$$\mathcal{L}_{\text{Adj-Match}}(\theta) = \sum_{t \in \{0,\dots,1-h\}} \left\| \frac{2}{\sigma(t)} \left( v_\theta^{\text{finetune}}(X_t, t) - v^{\text{base}}(X_t, t) \right) + \sigma(t) \, \tilde{a}_t \right\|^2. \tag{}$$

8:      Compute the gradient $\nabla_\theta \mathcal{L}(\theta)$ and update $\theta$ using a gradient descent step

9: **end for**

**Output:** Fine-tuned vector field $u^{\text{finetune}}$

---

## F  REWARD-GUIDED FLOW MERGING (RFM): COMPUTATIONAL COMPLEXITY, COST, AND APPROXIMATE FINE-TUNING ORACLES

Reward-Guided Flow Merging (RFM, see Alg. 1) is a sequential fine-tuning scheme which, at each of the ($K$) outer iterations, calls a reward-guided fine-tuning oracle such as REWARDGUIDEDFINE-TUNINGSOLVER (see Apx. E.2). In practice, each oracle call performs ($N$) gradient steps of Adjoint Matching (see Apx. E.2). At first sight, this suggests that the computational complexity of RFM scales linearly in $K$ with respect to a standard fine-tuning run with ($N$) steps. However, this worst-case view does not fully capture the practical computational cost. We highlight two observations.

**Approximate fine-tuning oracle.**  First, RFM can operate reliably with a rather *approximate fine-tuning oracle*, i.e., with relatively small values of ($N$). We evaluate this phenomenon by replicating the objective curve of Fig. 2d with same parameters and setting, for three different configurations of ($K, N$) that keep the total budget ($K \cdot N = 300$) fixed but vary the outer (i.e., $K$) and inner (i.e., $N$) iteration counts:

- $K = 10, ; N = 30$
- $K = 15, ; N = 20$ (as in Fig. 2d)
- $K = 30, ; N = 10$

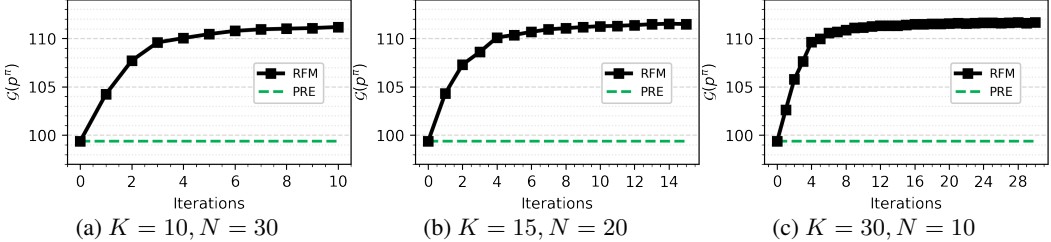

(a) $K = 10, N = 30$      (b) $K = 15, N = 20$      (c) $K = 30, N = 10$

Figure 6: (left) RFM run for reward-guided intersection with $K = 10, N = 30$, (center) RFM run for reward-guided intersection with $K = 15, N = 20$, (right) RFM run for reward-guided intersection with $K = 30, N = 10$.

The three corresponding curves are reported in Fig. 6. Empirically, all three settings achieve nearly identical final objective values, indicating that a more approximate oracle (smaller ($N$)) can be compensated by increasing the number of outer RFM iterations ($K$), and vice versa, as long as the total optimization budget remains comparable. We observe a similar behaviour also on real-world, higher-dimensional, experiments (see Sec. 7 and Apx. G.2), where we values of $K$ vary from $K = 1$ to $K = 37$.

$K/N$ **Trade-off.**  Second, the runtimes of these configurations are of the same order. On our implementation, the runs with $((K, N) = (10, 30), (15, 20), (30, 10))$ require approximately 1615 s, 1643 s, and 1870 s, respectively, showing a very light increase depending on $K$. This further supports the view that practitioners can trade off a cheaper but less accurate inner oracle (small ($N$)) against a slightly larger number of outer RFM steps (larger ($K$)), and vice versa, without incurring prohibitive additional cost. Since RFM effectively solves a convex/non-convex optimization problem in probability space, we believe that classic convex optimization provides an interpretable framework for trading-off $N$ and $K$, by interpreting $N$ as the typical step-size, or learning rate, and $K$ as the typical number of gradient steps. Clearly, higher learning rates typically require less gradient steps and vice versa. Ultimately, one should notice that increasing $N$ does not directly imply better solution quality of the fine-tuning oracle, as it is the case for the oracle we employ within Sec. 7 (i.e., Adjoint Matching (Domingo-Enrich et al., 2024)), for which performance can degrade for excessively high values of $N$.

## G EXPERIMENTAL DETAILS

### G.1 ILLUSTRATIVE EXAMPLES EXPERIMENTAL DETAILS

Numerical values in all plots shown within Sec. 7 are means computed over diverse runs of RFM via 5 different seeds. Error bars correspond to 95% Confidence Intervals.

**Shared experimental setup.** For all illustrative experiments we utilize Adjoint Matching (AM) [14 ] for the entropy-regularized fine-tuning solver in Algorithm 1. Moreover, the stochastic gradient steps within the AM scheme are performed via an Adam optimizer.

**Intersection Operator.** The balanced plot (see Fig. 2b is obtained by running RFM with $\alpha = [0.1, 0.1]$, for $K = 80$ iterations, $\gamma_k = 28$, and $\lambda_t = 0.2$ for $t > 1 - 0.05$, and $\lambda_t = 0.4$ otherwise.

For the balanced, reward-guided case in Fig. 2c, we consider a reward function that is maximized by increasing the $x_2$ coordinate. We run RFM with $\alpha = [0.1, 0.1]$, for $K = 15$ iterations, $\gamma_k = 1.2$, and $\lambda_t = 0.2$ for $t > 1 - 0.05$, and $\lambda_t = 0.4$ otherwise.

**Union Operator.**

In both cases, we learn a critic via standard f-GAN (Nowozin et al., 2016) with 300 gradient steps at each iteration $k \in [K]$ and continually fine-tune the same critic over subsequent iterations. For critic learning, we use a learning rate of $5 \exp(-5)$.

For the balanced case, in Fig. 2f, we run RFM with $\alpha = [1.0, 1.0]$. We use $K = 13$ iterations, $\gamma_k = 0.001$.

For the unbalanced case in Fig. 2g, we run RFM with $\alpha = [0.2, 1.8]$. Notice that up to normalization this is equivalent to $[0.1, 0.9]$ as reported in Fig. 2g for the sake of interpretability. We use $K = 13$ iterations, $\gamma_k = 0.001$.

**Interpolation Operator.** In both cases, we learn a critic via standard f-GAN (Nowozin et al., 2016) with 800 gradient steps at each iteration $k \in [K]$ and continually fine-tune the same critic over subsequent iterations. For critic learning, we use a learning rate of $1 \exp(-5)$, and gradient penalty of 10.0 to enforce 1-Lip. of the learned critic.

For the case where $\pi^{init} := \pi^{pre,1}$ (i.e., left pre-trained model), in Fig. 2j, we run RFM with $\alpha = [1.0, 1.0]$. We use $K = 6$ iterations, $\gamma_k = 1.0$.

For the case where $\pi^{init} := \pi^{pre,2}$ (i.e., right pre-trained model), in Fig. 2k, we run RFM with $\alpha = [1.0, 1.0]$. We use $K = 6$ iterations, $\gamma_k = 1.0$.

**Complex Logic Expressions via Generative Circuits.** Pre-trained flows $\pi_1$ and $\pi_2$, as well as $\pi_1$ and $\pi_2$ are intersected via RFM with $\gamma_k = 1$, for $K = 20$, and $\lambda_t = 0.1$. The union operator is implemented with $K = 30$, $\gamma_k = 0.0009$, 300 critic steps and learning rate $5 \exp(-5)$.

### G.2 MOLECULAR DESIGN CASE STUDY

Our base model FlowMol2 CTMC (i.e., PRE-1) (Dunn & Koes, 2024) is pretrained on the GEOM-Drugs dataset (Axelrod & Gomez-Bombarelli, 2022). We obtain our second model (i.e., PRE-2) by finetuning PRE-1 with AM (Domingo-Enrich et al., 2024) to generate poses with lower single point total energy wrt. the continuous atomic positions as calculated with dxtb at the GFN1-xTB level of theory Friede et al. (2024). We then run RFM with $K = 50$, $\gamma = 0.001$ for the balanced flow merging, and $K = 20$, $\gamma = 0.005$ to obtain the unbalanced flow merging. For reward-guided flow merging (RFM-RG), we set $\gamma = 0.1$ and obtain the best model after $K = 11$. All models start from PRE-1, i.e., $\pi^{init} = \pi^{pre,1}$. All results for merging pre-trained models on GEOM can be found in Table 1. Running RFM-RG with $\alpha = 3$ and $\gamma = 0.001$, we obtain a model after $K = 35$ that keeps the validity of its base models while implementing the reward-guided intersection. We note that beyond validity, a critical step towards practical application will be to integrate molecular stability and synthesizability. Our RFM formulation straightforwardly supports these extensions in the reward functional, and we leave their implementation to future work. For our second case-study - the OR operator - we use FlowMol2 CTMC pre-trained on QM9 (Ramakrishnan et al., 2014). We limit dimensionality to reduce the problem complexity by sampling 10 atoms per molecule, and run RFM with $\gamma = 100$, $K = 37$. In particular Figure 7 shows that the estimated mean of the model

| Model | Mean total energy [Ha] | Mean validity [%] |
|-------|------------------------|-------------------|
| PRE-1 | $-8.09 \pm 0.31$ | $76.44 \pm 1.7$ |
| RFM-B | $-10.95 \pm 0.28$ | $74.34 \pm 0.9$ |
| RFM-RG | $-12.85 \pm 0.16$ | $74.02 \pm 1.18$ |
| RFM-UB | $-13.69 \pm 0.28$ | $72.78 \pm 0.4$ |
| PRE-2 | $-14.76 \pm 0.29$ | $68.04 \pm 0.8$ |

Table 1: Mean total energy and validity with standard deviation, averaged over 5 different seeds. Suffixes: B - balanced ; UB - unbalanced; RG - reward-guided flow merging

$\pi^*$ obtained via RFM matches the average total energy of $\pi^{pre,1}$ and $\pi^{pre,2}$ as predicted by the closed-form solution for the union operator presented in Sec. 3. In Fig. 7, OR denotes the final policy $\pi^*$ returned by RFM.

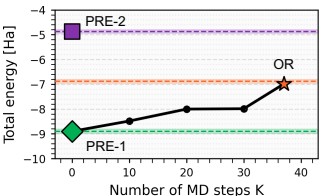

Figure 7: Union on QM9

### G.3 CONFORMER GENERATION CASE STUDY

We finetune the GEOM-QM9 pre-trained ETFlow model (denoted PRE-1) with AM on the molecular system C#C[C@H](C=O)CCC to obtain PRE-2, using the same total energy objective as in the molecular design case study. This is also the molecular system we perform our evaluations on. For the subsequent merging experiments, we choose the lower-energy PRE-2 as the base model, i.e., $\pi^{init} = \pi^{pre,2}$. Balanced merging is performed with $\alpha_1 = \alpha_2 = 1$, $\gamma = 0.025$ and $K = 6$. The unbalanced merging is run with $\alpha_1 = 0.7$ and $\alpha_2 = 0.3$ and we take the model after $K = 8$ steps with $\gamma = 5e-5$. The reward-guided merging model was obtained with $\gamma = 0.025$ after $K = 6$, and the union model after $K = 1$ with $\gamma = 1e-3$ and critics with the same GNN backbone as ETFlow. We show all results for the conformer geneation case study in Tab. 2

| Model | $E$ [kcal/mol] | $\mu$ [debye] | $\Delta\epsilon$ [kcal/mol] | $E_{min}$ [kcal/mol] |
|-------|----------------|---------------|------------------------------|----------------------|
| PRE-1 | $0.3385 \pm 0.0002$ | $0.1679 \pm 0.0002$ | $0.5373 \pm 0.0019$ | $0.2793$ |
| RFM-UB | $0.3412$ | $0.1512$ | $0.5173$ | $0.2778$ |
| RFM-B | $0.3356 \pm 0.0001$ | $0.1503 \pm 0.0002$ | $0.4915 \pm 0.0014$ | $0.2782$ |
| RFM-UNION | $0.3352$ | $0.1467$ | $0.5033$ | $0.2761$ |
| RFM-RG | $0.3193 \pm 0.0003$ | $0.1141 \pm 0.0002$ | $0.4849 \pm 0.0015$ | $0.2777 \pm 0.0008$ |
| PRE-2 | $0.3175 \pm 0.0006$ | $0.1268 \pm 0.0006$ | $0.4819 \pm 0.0010$ | $0.2761 \pm 0.0027$ |

Table 2: Median Absolute Errors for energy $E$, dipole moment $\mu$, HOMO-LUMO gap $\Delta\epsilon$, and minimum energy $E_{min}$ across different models. We report mean and standard deviation over 5 different seeds.
Suffixes: RG - reward-guided flow merging

# H BEYOND MOLECULES: REWARD-GUIDED FLOW MERGING OF PRE-TRAINED IMAGE MODELS

We further showcase the capabilities of Reward-Guided Flow Merging on a small-scale, yet informative experiment for image generation. In the following, we consider pretrained CIFAR-10 image models (Krizhevsky et al., 2009) and use the LAION aesthetics predictor V1 (Schuhmann et al., 2022) as a reward model. Specifically, the aesthetics predictor was trained on a subset of the SAC dataset (Pressman et al., 2022) with available ratings from 1 (low preference / aesthetics) to 10 (high preference). The goal of this case study is to show that RFM can merge two models, PRE-1 and PRE-2, while optimizing the aesthetics score. We perform reward-guided flow merging with PRE-2 as the base model, obtaining the model RFM-RG after $K = 11$ iterations with $\gamma = 1$ and $\alpha_i = 1$. The numerical results in Tab. 3 show that RFM can successfully intersect multiple prior flow image models while maximizing the aesthetic score. In particular, the fine-tuned model achieves a score of $3.64 \pm 0.53$ against $3.16 \pm 0.66$ and $3.23 \pm 0.58$ of PRE-1 and PRE-2 respectively. We also report sample images of the discussed models in Fig. 8.

| Model | Mean aesthetic score |
|-------|----------------------|
| PRE-1 | $3.16 \pm 0.66$ |
| PRE-2 | $3.23 \pm 0.58$ |
| RFM-RG | $3.64 \pm 0.53$ |

Table 3: RFM can perform reward-guided (RG) intersections of pre-trained CIFAR-10 image models (Krizhevsky et al., 2009). We evaluate the resulting models in terms of mean aesthetic score (i.e., the reward) over 1000 samples, and report one std.

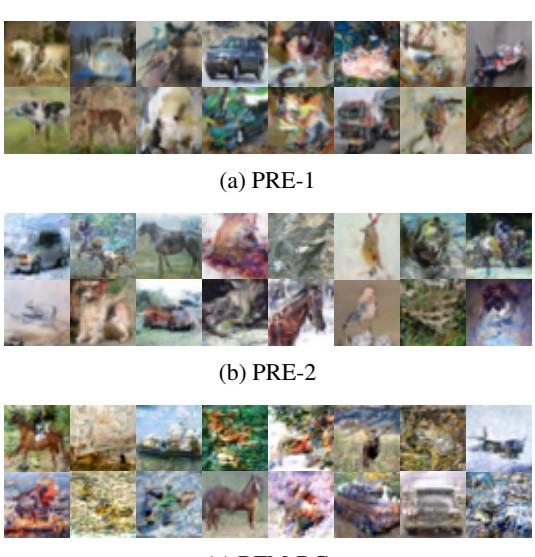

(a) PRE-1

(b) PRE-2

(c) RFM-RG

Figure 8: Images generated by the two pre-trained flow models (i.e., PRE-1, PRE-2), and by the flow model obtained via reward-guided intersection (i.e., RFM-RG).

