# OpenReview forum: "Reward-Guided Flow Merging via Implicit Density Operators"
_ICLR.cc/2026/Conference — Submitted to ICLR 2026_

### Official Review · Reviewer_vWQC · 2025-10-20

**Soundness:** 2
**Presentation:** 4
**Contribution:** 3
**Rating:** 4
**Confidence:** 3

**Summary:**

My expertise is in AI for Science, not core AI theory. Therefore, this review primarily assesses the paper's practical application, experimental validation, and its claimed contribution to scientific tasks. I have focused less on the fundamental methodological novelty in the pure AI domain, deferring that judgment to other reviewers.

This paper presents a novel, theoretically-grounded framework for unifying reward-guided optimization with flow model merging. The concept is well-motivated and clearly demonstrated on multiple well-structured toy examples.

However, the empirical validation on the high-dimensional task is limited. There is only one application task. Moreover, the core reward-based optimization (RFM-RB) fails, performing significantly worse than a simpler baseline in both key metrics (energy and validity).
These results restrict the method's practical utility.

In summary, while the paper introduces a promising and theoretically sound framework, its practical effectiveness is not yet thoroughly validated due to limited experimental results on high-dimensional tasks.

---
**The usage of LLM**: I wrote the entire review myself and only used the LLM to correct the grammar and improve readability.

**Strengths:**

1. **Unified Framework:** A primary strength is the novel formulation that unifies reward-guided optimization and flow merging into a single objective function. This framework is supported by a solid theoretical background and demonstrated with diverse operators and toy examples (e.g., intersection, union, interpolation).
2. **Clear Motivation and Methodology:** The paper presents a clear motivation and a well-articulated methodology. This is effectively explained in Sections 3 and 4 and well-illustrated through several intuitive toy examples.
3. **Demonstrated Feasibility on High-Dimensional Data:** The authors successfully demonstrate the feasibility of their flow merging approach on a high-dimensional task (molecular generation) that involves both discrete (molecular graph) and continuous (3D coordinates) data.

**Weaknesses:**

The paper effectively demonstrates its motivation and capabilities through several well-designed toy examples. However, its validation on high-dimensional tasks appears insufficient.

**1. Significant performance degradation on optimization setting**

A core claim of the paper is to _simultaneously_ optimize a reward function while merging multiple flows.
> **Page 1, Line 46**: *Can we fine-tune a pretrained flow model to optimize a given reward function while integrating information from (i.e., merge) multiple pre-trained flows.*

However, the empirical evidence for the reward-guided optimization component is weak and concerning.
- The primary optimization experiment (RFM-RB) is not presented in the main manuscript and is only available in Appendix F.2.
- The results in Table 1 show that RFM-RB fails to outperform the PRE-2 baseline (which was trained with standard Adjoint Matching). In fact, RFM-RB performs significantly worse: it achieves a worse mean total energy (−12.47 Ha) compared to the baseline while suffering a significant drop in molecular validity (from 76% to 33%).

The authors attribute this performance degradation to the multi-objective nature of molecular design.
However, this explanation is insufficient, as the adjoint matching (PRE-2) demonstrates a better trade-off, improving energy significantly (−14.76 Ha) with a minor validity drop (from 76% to 68%).

This poor performance undermines the paper's central claim.
The authors must provide additional evidence to validate that RFM can practically solve reward-based optimization problems beyond its theoretical guarantees.

**2. Limited Application Study**

The paper is limited to a single application: a molecular discovery task.
This contrasts with other RL-based finetuning methods, which are often validated on well-established tasks (e.g., image generation) with diverse baselines[1] or across multiple domains[2,3].


**3. Ambiguity of experiment name**

There is a significant ambiguity in the naming of the high-dimensional experiment that is likely to mislead domain experts.
The authors repeatedly describe the task as "conformer generation" including in the abstract.

However, this task is **3D molecular generation** (i.e., jointly generating the molecular graph and 3D coordinates).
This is fundamentally different from **molecular conformer generation**, which involves generating 3D coordinates for _given_ molecule(s).

This misnaming caused significant confusion. I strongly recommend the authors correct this terminology throughout the manuscript to accurately reflect the experimental task.
- **3D molecular generation:** Joint design of molecular graph (categorical) and its 3D conformer (continuous). (e.g., FlowMol[4], SemlaFlow[5], CGFlow[6])
- **Molecular conformer generation:** 3D conformer generation of a _given_ molecule. (e.g., Torsional Diffusion[7], ETFlow[8])

---
**Reference**
1. Domingo-Enrich, Carles, et al. "Adjoint matching: Fine-tuning flow and diffusion generative models with memoryless stochastic optimal control." _arXiv preprint arXiv:2409.08861_(2024).
2. Venkatraman, Siddarth, et al. "Amortizing intractable inference in diffusion models for vision, language, and control." _Advances in neural information processing systems_ 37 (2024): 76080-76114.
3. Venkatraman, Siddarth, et al. "Outsourced diffusion sampling: Efficient posterior inference in latent spaces of generative models." _arXiv preprint arXiv:2502.06999_ (2025).
4. Dunn, Ian, and David Ryan Koes. "Mixed continuous and categorical flow matching for 3d de novo molecule generation." _ArXiv_ (2024): arXiv-2404.
5. Irwin, Ross, et al. "SemlaFlow--Efficient 3D Molecular Generation with Latent Attention and Equivariant Flow Matching." _arXiv preprint arXiv:2406.07266_ (2024).
6. Shen, Tony, et al. "Compositional Flows for 3D Molecule and Synthesis Pathway Co-design." _arXiv preprint arXiv:2504.08051_ (2025).
7. Jing, Bowen, et al. "Torsional diffusion for molecular conformer generation." _Advances in neural information processing systems_ 35 (2022): 24240-24253.
8. Hassan, Majdi, et al. "Et-flow: Equivariant flow-matching for molecular conformer generation." _Advances in Neural Information Processing Systems_ 37 (2024): 128798-128824.

**Questions:**

- What do the authors perceive as the primary limitations of the proposed RFM approach?

- Questions related to the molecular design task:
    - **Figure 3 and Table 1**: What do `RFM-B` and `RFM-UB` stand for? I assume those mean "balanced" and "unbalanced," which are mentioned in the text, but this is not explicitly defined in the figure/table captions.
    - **Page 23 Line 1231**: What reward function was used to obtain the PRE-2 model via AM? Was the exact same reward function used for the RFM-RB experiment?
    - What was the initial model ($\pi_\text{init}$) used for the RFM-B, RFM-UB, and RFM-RB experiments? Was it PRE-1 or PRE-2?

- Minor Typos/Formatting:
	- **Page 2, Line 82**: Could you change the citation format from "$p_{data}\text{Lipman et al. ...}$" to "$p_{data}~(\text{Lipman et al. ...})$"?
	- **Page 5, Line 236**: The hyperlink for the equation 8 appears to be missing.
    - **Page 22, Line 1163 (in Algorithm 3, step 7)**: The objective function is referred to as '??'. MOreover, the equation number is missing.

---

> ### Author Response · Authors · 2025-11-23
>
> We thank the Reviewer for considering our framework novel, well-motivated, theoretically sound, and promising, as well as the method well-articulated and illustrated. Within the updated version of the paper, we have used one extra available page (according to ICLR guidelines) to significantly expand and strengthen the experimental evaluation of the proposed method according to the Reviewer suggestions. In the following, we aim to sharply address the points raised by the Reviewer.
>
> **Significant performance degradation on optimization setting**
>
> We thank the Reviewer for raising this point and fully agree with their observations. We have updated the paper presenting results for reward-guided flow merging, which we renamed as RFM-RG, within the main paper (Sec. 7) both for the molecular design task mentioned by the Reviewer (see RFM-RG in Fig. 3.h), as well as in a new conformer generation task, which we mention within the replies below (see RFM-RG in Fig. 5) and introduce within the paragraph "Flow Merging of Conformer Generation Models". With respect to the experimental result mentioned by the Reviewer, in Fig. 3.a we now show the validity-energy of the initial flow models PRE-1 and PRE-2 and report RFM-B, which is obtained via (pure, i.e., reward agnostic) flow merging, by minimization of KL divergences from the prior models PRE-1 and PRE-2. We then report the model obtained via reward-guided flow merging (RFM-RG), where an energy minimization term is added to the merging formulation according to Eq. 5. As one can notice, RFM-RG achieves lower energy than RFM-B (i.e., -12.85 vs  -10.95), which is obtained via pure merging, while preserving similar validity (i.e., 74.02 vs 76.44). In particular, the significant drop in validity within the previous version of the manuscript was due to improper parametrization of the scheme. Analogous positive results for reward-guided flow merging are shown in Fig.5, where we evaluate energy (a), dipole moment (b), HOMO-LUMO gap (c), and min. energy (d) in a conformer generation task. As one can see (e.g., in Fig. 5.a), RFM-RG achieves significantly lower energy values (i.e., higher reward) than RFM-B. These results, added within the updated version of the manuscript, showcase the ability of RFM to successfully perform reward-guided flow merging.
>
> Ultimately, we wish to point out that, to our understanding, the Reviewer seems to consider Adjoint Matching (AM) as a possible baseline. However, this is conceptually wrong for two fundamental reasons: (1) AM tackles reward-guided fine-tuning, but it cannot solve (or be used for) the problem we are considering in this work, where multiple prior models are available, (2) for the sub-case of our framework that can be tackled by AM, the closed-form solution of RFM is exactly identical to that of AM. In other words, for the case of one prior model (or $\alpha_i = 0$ for all prior models except one), our framework retrieves exactly the same solution as the one induced by the AM method, which corresponds to the reward-tilted distribution (see e.g., [1, Eq. 1]).
>
> **Limited Application Study**
> We agree with the Reviewer that the previous version of the work was somewhat limited in terms of application studies. To this end, we used the extra available page to extend the experimental section of the work by adding an evaluation of our framework on molecule conformer generation models based on ETFlow [2].
> The Reviewer can find a new paragraph named "Flow Merging of Conformer Generation Models" within Sec. 7 of the updated paper and related experimental results within Fig. 5. These results further showcase the ability of RFM to perform balanced (B), unbalanced (UB), reward-guided (RG) intersections, as well as unions (UNION) of prior ETFlow [1] conformer generation models for a specific molecular system. We evaluate the resulting flow models in terms of energy (5a), dipole moment (5b), HOMO–LUMO gap (5c), and minimum energy (5d). Although we do not demonstrate a full real-world deployment of our method, we believe that the updated experimental section of the paper showcases strong and promising performance for a fundamentally new, theory-backed algorithm, that we view as the central contribution of this work, which is primarily of mathematical and algorithmic nature.
>
>
> **Ambiguity of experiment name**
> We thank the Reviewer for pointing this out and we have updated the experiment name, as well as introduce another experimental evaluation on molecular conformer generation based on ETFlow [2] within Sec. 7 of the updated manuscript.

---

> > ### Author Response · Authors · 2025-11-23
> >
> > **Questions**
> >
> > 1. (Limitations of RFM) Given that RFM is a particularly novel method, as it relies on probability-space optimization rather than only standard RL machinery, we believe it might need further improvements to close the gap with real-world impact. On the other hand, from an evaluation viewpoint, we found particularly challenging to be able to evaluate the obtained generative models in a meaningful way. We believe that extensions of this work towards real-world applications would make it possible to overcome both mentioned limitations of this work.
> >
> > 2. (Typos) We thank the Reviewer for reporting these typos. We have fixed them within the updated version of the work, with the exception of Eq. 8, which is an equation within the referenced paper rather than within our work.
> >
> > 3. (Experimental details) We have updated the work with updated details within Appendix G.2. We thank the Reviewer for mentioning these points.
> >
> >
> > **References**
> >
> > [1] Adjoint Matching: Fine-tuning Flow and Diffusion Generative Models with Memoryless Stochastic Optimal Control, Carles Domingo-Enrich, 2024.
> >
> > [2] Et-flow: Equivariant flow-matching for molecular conformer generation, Hassan Majdi, 2024.

---

> > > ### Comment · Reviewer_vWQC · 2025-11-26
> > >
> > > Thank you for response.
> > >
> > > However, There are some concerns not yet addressed.
> > >
> > > **Significant performance degradation on optimization setting**
> > >
> > > I agree that RFM don't have to outperform the Adjoint matching to be accepted in ICLR.
> > >
> > > I would like to clarify my theoretical concern regarding the trade-off, which I feel was not fully resolved by the initial explanation of "multi-objective nature.":
> > > > We attribute this to the multi-objective nature of molecular design: the single-objective reward in our case-study does not penalize invalid molecules.
> > >
> > > If we consider the trade-off between energy and validity, the performance of the prior models (PRE-1 and PRE-2) essentially establishes a reference for the Pareto frontier. In the initial manuscript, the validity dropped significantly below both priors, which suggested a failure in the flow merging mechanics rather than just a trade-off characteristic.
> > >
> > > In the revised manuscript, updated model (RFM-RG)'s performance looks good.
> > >
> > > **Limited Application Study**
> > >
> > > Thank you for the addition of the "Flow Merging of Conformer Generation Models" experiment.
> > >
> > > However, I believe the target audience of this paper is general audience in AI domain. Current applications are limited to the molecular domain, Therefore, it is required to test on different domains over the single molecular domain.

---

> > > > ### Author Response · Authors · 2025-12-03
> > > >
> > > > We thank the Reviewer for their reply and recognizing that we solved the concern regarding performance degradation. In the following we further address the points raised.
> > > >
> > > > **Significant performance degradation on optimization setting**
> > > >
> > > > We believe the updated experiments showcase positive results of our method in a convincing way, and we thank the Reviewer for acknowledging that our method performance within the updated version 'looks good'.
> > > >
> > > > Ultimately, we wish to point out that the sentence "RFM doesn't have to outperform the Adjoint matching to be accepted in ICLR" does not make logical sense. As explained in our previous reply, RFM and AM solve different problems and therefore cannot be neither practically nor theoretically compared. More concretely, RFM is built on top of AM, and would collapse to AM to tackle the trivial problem instance where only one prior model is considered and the KL divergence is chosen as regularizer for reward-guided fine-tuning. On the other hand, AM would be able to solve only a linearization of the problem tackled by RFM. This would correspond exactly to performing one iteration of RFM, which is both theoretically and experimentally (see Sec. 7) strongly sub-optimal compared against RFM with $K>1$.
> > > >
> > > >
> > > > **Limited Application Study**
> > > >
> > > > The presented framework and method has been developed in strong collaboration with a computational chemistry group, and we are confident that the proposed method is highly relevant, and potentially significantly impactful for researchers working at the intersection of generative models and molecular design, as well as related discovery problems over chemical spaces, which are highly practically relevant. Nonetheless, as asked by the Reviewer, we conducted an experimental analysis of the proposed method on the image domain, and report an overview and results within Appendix H (page 27, orange text) titled "Beyond Molecules: Reward-Guided Flow Merging of Pre-Trained Image Models". To summarize, we show that RFM can successfully perform reward-guided flow merging in a small-scale, yet informative setting. In particular, we show that it can maximize the aesthetic score of the fine-tuned model while preserving information from multiple prior models. Moreover, compared with the molecular experiments, in this case we show that the fine-tuned model can achieve a mean reward even higher than both prior models. Ultimately, we wish to point out that RFM is to our knowledge the first method that can solve this problem both in practice and with theoretical guarantees.

---

### Official Review · Reviewer_sG1a · 2025-11-01

**Soundness:** 3
**Presentation:** 3
**Contribution:** 3
**Rating:** 8
**Confidence:** 2

**Summary:**

This paper introduces Reward-Guided Flow Merging (RFM), a unified framework that jointly addresses reward-guided adaptation of pre-trained flows and integration of multiple models. The method formulates merging as optimization over diverse implicit density operators, such as intersection, union, and interpolation. RFM employs a mirror-descent scheme that converts complex merging tasks into sequential fine-tuning problems. The paper presents theoretical proofs as well as experiments on the drug design task.

**Strengths:**

The paper presents a unified theoretical framework that generalizes two difficult problems in AI for science, reward-guided fine-tuning and model merging. This framework will benefit the design of many scientific models. The paper also provides rigorous proofs of theoretical guarantees for the proposed Reward-Guided Flow Merging algorithm, providing a reliable foundation for the framework.

**Weaknesses:**

The paper only evaluates its framework on the molecular design task. This limits the demonstration of the claimed wide real-world applications. More experiments in future works would strengthen the generality and practical impact of the algorithm.

**Questions:**

The paper provides an effective framework for merging pre-trained models. Could the authors discuss more about how model performance and efficiency scale as the number of merged models increases?

---

> ### Author Response · Authors · 2025-11-23
>
> We thank the Reviewer for recognizing our framework as unifying two difficult problems in AI for science, effective, and with reliable theoretical foundations. In the following, we aim to sharply address the points raised by the Reviewer.
>
> **Limited experiments**
>
> We agree with the Reviewer that the previous version of the work was somewhat limited in terms of real-world experimental evaluation of the proposed method. To this end, we used one extra available page to expand the experimental section of the work by adding an evaluation of our framework on another highly-relevant scientific discovery task, namely graph conditioned molecular conformer generation, based on the recent ETFlow model [1]. The Reviewer can find a new paragraph named "Flow Merging of Conformer Generation Models" within Sec. 7 of the updated paper and related experimental results within Fig. 5. These results further showcase the ability of RFM to perform balanced (B), unbalanced (UB), reward-guided (RG) intersections, as well as unions (UNION) of prior ETFlow [1] conformer generation models for a specific molecular system. We evaluate the resulting flow models in terms of energy (5a), dipole moment (5b), HOMO–LUMO gap (5c), and minimum energy (5d). Given the new experimental evaluation with positive results, we believe that the updated experimental section showcases strong and promising performance for a fundamentally new, theory-backed algorithm, that we view as the central contribution of this work, which is primarily of mathematical and algorithmic nature.
>
> **Scaling complexity w.r.t. number of prior models**
>
> Since the model fine-tuned is always only one, as one can notice by inspecting Alg. 1, the only algorithmic dependence on the number of prior models lies in the estimation of the first variation $\delta \mathcal{G}$ (line 4, Alg. 1). Crucially, its definition depends on the specific operator that one wishes to implement, and therefore also its dependence on the number of prior models. In particular, we can identify 3 representative cases:
>
> 1. (Closed-form  gradient of first variation) Since certain operators can be expressed via KL divergences (e.g., the intersection, or reward-guided intersection operator, see Eq. 6), which admit closed-form expressions for the gradient of their first variation, as presented within Sec. 4, these would lead to a computational complexity seemingly independent w.r.t. the number of prior models.
>
> 2. (Gradients of first variations via one learned critic) As proved in Proposition 1 and explained within the paragraph "Implementation of Intersection, Union, and Interpolation operators.", certain operators (e.g., union) can be re-written such that it is sufficient to estimate only one gradient rather than $n$ many, where $n$ is the number of prior models (see Eq. 14). As in the previous case, this would lead to a computational complexity seemingly independent w.r.t. the number of prior models.
>
> 3. (Non-trivial dependency) Due to the generality of the formulation in Eq. 5, there can be operators such that the estimation of gradients incurs in a complexity scaling linearly with the number of prior models. At the moment, we do not visualize a case where the complexity would provably scale with a worse rate, but this might be possible for certain operators. Nonetheless, observations such as the one provided in Proposition 6 might be used to render the method more scalable by breaking the dependency of computational complexity on the number of prior models.
>
> **References**
>
> [1] Et-flow: Equivariant flow-matching for molecular conformer generation, Hassan Majdi, 2024.

---

### Official Review · Reviewer_6Mtg · 2025-11-01

**Soundness:** 1
**Presentation:** 3
**Contribution:** 1
**Rating:** 2
**Confidence:** 4

**Summary:**

This work proposes a generative optimisation framework that advocates task-aware reward-guided adaptation of multiple pretrained flow models. The proposed formulation entails implicit density operators (union, intersection, interpolation, and their combinations) over generative model densities. The authors also implement a mirror-descent scheme to approximate the objective in terms of a sequence of reward-guided fine-tuning problems.

nprecedented progress in large-scale flow and diffusion modeling for scientific discovery recently raised two fundamental challenges:  reward-guided adaptation of pre-trained flows, and integration of multiple models, i.e., model merging. While current approaches address them separately, we introduce a unifying probability-space framework that subsumes both as limit cases, and enables reward-guided flow merging. This captures generative optimization tasks requiring information from multiple pre-trained flows, as well as task-aware flow merging (e.g., for maximization of drug-discovery utilities). Our formulation renders possible to express a rich family of implicit operators over generative models densities, including intersection (e.g., to enforce safety), union (e.g., to compose diverse models) and interpolation (e.g., for discovery in data-scarce regions). Moreover, it allows to compute complex logic expressions via generative circuits. Next, we introduce Reward-Guided Flow Merging (RFM), a theory-backed mirror-descent scheme that reduces reward-guided flow merging to a sequential fine-tuning problem that can be tackled via scalable, established methods. Then, we provide first-of-their-kind theoretical guarantees for reward-guided and pure flow merging via RFM. Ultimately, we showcase the capabilities of the proposed method on illustrative settings providing visually interpretable insights, and on a high-dimensional drug design task generating low-energy molecular conformers.

**Strengths:**

--- The manuscript is generally easy-to-read.

--- Model composition and fine-tuning are clearly, and justifiably, extremely active areas of research, so the work deals with a topical subject.

**Weaknesses:**

I apologise in advance for what might come across as a rather disappointing/critical review for the authors, but I put in extensive effort on reviewing this paper to the best of my ability and knowledge of the field.

— What the authors call 'merging' is typically referred to as composition in the literature. An entire body of work has been dedicated to composition  (including under constraints and/or using density operators including based on logical And-Or-Not Operators, Ito operators, Feynman-Kac etc.) that has been completely sidestepped in the current work. See, e.g., references [1, 2, 4, 5, 6, 7, 8, 9]. Not only should the work have been contextualised in terms of similarities and differences with this prior work, but comprehensive empirical benefits over them should have been shown.

There are other works such as [3] which being contemporaneous are not subject to this criticism.

-- For a technical/optimisation perspective, "reward guidance" is not central to the work and comes across as rather peripheral/contrived since the solution to the formulation only cares about the convexity/concavity of the problem and availability of the function gradient (including, or without, the reward). Similarly, the generality of "merging" in the current context is overstated. As the formulation is restricted to an affine/convex combination of divergences, it cannot implement important operations such as negation and contrast. As soon as the overall concavity/convexity is violated, no global convergence guarantee holds as the classical stochastic approximation theory (Robbins-Monro style updates) can only guarantee a local solution.

--- The entire framework is essentially a straightforward amalgamation of existing ideas, and it's unclear how this work advances the field at all. In particular, heavily derives from/relies on key notions and machinery already tackled in the literature on RL under general utilities and stochastic optimal control [10, 11, 12, 13].

--- Experiments are also underwhelming, with essentially no meaningful comparisons included against state-of-the-art baselines on composition and fine-tuning.

[1] Khalafi et al. Constrained Diffusion Models via Dual Training. NeurIPS 2024.

[2] Giannone et al. Aligning Optimization Trajectories with Diffusion Models for Constrained Design Generation. NeurIPS 2023.

[3] Khalafi et al. Composition and Alignment of Diffusion Models using Constrained Learning. arXiv 2025.

[4] Garipov et al. Compositional Sculpting of Iterative Generative Processes. NeurIPS 2023.

[5] Thornton et al.  Composition and Control with Distilled Energy Diffusion Models and Sequential Monte Carlo. AISTATS 2025.

[6] Skreta et al. The superposition of Diffusion Models using the Ito Density Estimator. ICLR 2025.

[7] Skreta et al. Feynman-Kac Correctors in Diffusion: Annealing, Guidance, and Product of Experts. ICML 2025.

[8] Karczewski et al. Devil is in the Details: Density Guidance for Detail-Aware Generation with Flow Models. ICML 2025.

[9] Shih et al. Long Horizon Temperature Scaling. ICML 2023.

[10] Zhang et al. Variational Policy Gradient Method for Reinforcement Learning with General Utilities. NeurIPS 2020.

[11] Domingo-Enrich et al.  Adjoint Matching: Fine-tuning Flow and Diffusion Generative Models with Memoryless Stochastic Optimal Control. arXiv 2024.

[12] Han et al. Stochastic Control for Fine-tuning Diffusion Models: Optimality, Regularity, and Convergence. ICML 2025.

[13] Uehara et al. Fine-tuning of continuous-time diffusion models as entropy-regularized control. arXiv 2024.

**Questions:**

Could you please address my concerns detailed in the weaknesses section? In addition, wondering

(1) what the computational cost of the entire procedure is (noting that a sequence of fine-tuning steps needs to be solved)?

(2) what the effect of inexact updates is in practice in terms of discrepancy from the solution arrived through exact updates?

---

> ### Author Response · Authors · 2025-11-23
>
> We thank the Reviewer for the time and effort dedicated to evaluating our work, and considering the treated problem a topical subject. However, we believe that the review is based on severe factual inaccuracies and misunderstandings of our contributions. In the following, we aim to sharply and carefully clarify these points.
>
>
> **Model composition**
>
> The model composition literature typically deals with the problem of inference-time composition. On the contrary, our paper tackles the problem of model merging via fine-tuning. Crucially:
>
> 1. The two problems (i.e., composition at inference/sampling-time and model merging [e.g., 14]) are completely different from a practical and algorithmic standpoint. Concretely, model merging allows to discard the initial pre-trained models after the merging operation, while inference-time adaptation requires permanently storing all pre-trained models. Model merging operations are computed only one time to 'fuse' models, inference-time compositions are performed at sampling time, typically leading to significant per-sample computational cost. All papers mentioned by the Reviewers, namely  [1, 2, 4, 5, 6, 7, 8, 9], are either not directly related to composition/merging [i.e., 1, 2, 8, 9], or tackle inference-time model composition [i.e., 4, 5, 6, 7], which, as discussed, is a fundamentally different setting than the one treated in our work, i.e., (reward-guided) model merging.
>
> 2. Within the original submission we already discussed the relations of our work with the model composition literature within the second paragraph of the Related Works section, which we have now renamed "Diffusion and flow model merging and inference-time composition" without altering its content. In particular, we compared our framework with the work most related to ours, which is [6] (as mentioned by the Reviewer), as it formalizes the problem of inference-time model composition via a probability-space, or density, viewpoint. In comparison, our work introduces a completely different method, which allows to transport a similar probability-space and operator viewpoint from inference-time composition to a model merging setting.
>
>
> **Concurrent Work**
>
> The (extremely) concurrent work [3] mentioned by the Reviewer is the only reference mentioned, which tackles a problem certainly related to ours, further showing the relevance and timeliness of our contribution. Nonetheless, [3] deals with a different problem, and proposes a substantially different method, theoretical guarantees, and experimental evaluation. In particular, [3]:
>
> 1.  Does not tackle the reward-guided model merging problem, which is central in our work.
>
> 2. Seems to consider a significantly more restrictive class of divergences.
>
> 3. Presents a fundamentally different algorithmic scheme, which in particular, does not rely on calculus of variations.
>
> 4. Presents a profoundly different type of theoretical analysis and guarantees.
>
> In conclusion, while being a concurrent work, we agree that [3] deals with a problem highly related to ours, although it provides fundamentally different contributions (i.e., mathematical framework, algorithmic machinery, theoretical guarantees). In conclusion, to our knowledge these two concurrent works ([3] and ours) are the first tackling a highly relevant and timely problem, while presenting significantly different contributions.
>
> **Reward guidance is not central**
>
> We are somewhat confused by this point and not sure about what the Reviewer wishes to express. The proposed algorithm reduces (reward-guided) model merging to a sequence of reward-guided fine-tuning steps, with (surrogate) reward functions automatically computed via the use of calculus of variations (i.e., the reward function corresponds to a first variation, or functional gradient, as presented in Sec. 4). This effectively reduces complex (reward-guided) model merging operations to standard reward-guided fine-tuning, which enjoys a very rich literature, both for theory (e.g., [15]) and scalable methods (e.g., [11]).

---

> > ### Author Response · Authors · 2025-11-23
> >
> > **Limited expressivity and need of convexity**
> >
> > 1. (Need of convexity) Within the work we proposed a representative sample of highly-relevant density operators (i.e., intersection, union, interpolation, and their reward-guided counterparts). Nonetheless, the formulation proposed is neither tied to such set of operators, nor limited to convex divergences/functionals. In fact, convexity is a property which we leverage only to provide strong (global convergence) theoretical guarantees, but concretely it is not required, as it is the case in the vast majority of ML methods, where neural networks are used as function approximators. In conclusion, our mathematical framework, presented in Sec. 3, does not require convexity.
> >
> > 2. (Limited expressivity) While this work indeed proposes an objective corresponding to an affine combination of divergences, the vast majority of the presented contributions (i.e., method, theory) straightforwardly extend to more complex functional expressions. For instance, the proposed framework immediately generalizes to include inverse divergences (i.e., similarity measures) within the mentioned affine combination. This would directly allow to implement a 'subtraction' operator as mentioned by the Reviewer, e.g., by considering a similarity measure defined by $Q(p,q) = - KL(p,q)$, corresponding to simply adding a minus sign.
> >
> >
> > **Framework is a straightforward amalgamation of existing ideas**
> > The presented framework introduces a first-of-its-kind probability space viewpoint on reward-guided model merging, a problem neither formalized in prior work, nor solvable with existing methods, and leverages machinery from calculus of variations, which is strictly more general than standard RL methods, to tackle this general class of problems in a truly novel manner.  Concretely, this work advances the field by providing:
> >
> > 1. A fundamentally new and unified probability-space optimization viewpoint on reward-guided flow merging, which retrieves reward-guided fine-tuning and model merging as limit sub-cases.
> >
> > 2. The first principled and truly scalable method to compute the flow/diffusion model corresponding to the intersection of multiple prior models via sequential fine-tuning.
> >
> > 3. The first principled and truly scalable manner to compute reward-guided model merging, which generalizes reward-guided fine-tuning of diffusion/flow models to the case of divergence regularization with multiple prior models.
> >
> > 4. First-of-their-kind theoretical guarantees for reward-guided flow merging, retrieving highly relevant and novel guarantees for (pure) model merging as a sub-case. To our knowledge, these are among the first theoretical guarantees for diffusion/flow model merging.
> >
> > We believe that such contributions are profound, highly novel, and relevant both from a theoretical and practical standpoint. Ultimately, we wish to point out that a significant part of the contributions of this work (e.g., framework, theory, method) straightforwardly extend to other highly-relevant classes of generative models, including LLMs, and certain  discrete diffusion models [e.g., 16]. This is due to the fact that such models allow for direct estimation of the first variations required by our method.
> >
> > **Relation to General Utilities RL**
> > General Utilities or Convex RL [e.g., 10] is a sub-field of RL, where an agent wishes to compute a policy $\pi$ such that it maximizes a distributional (convex) objective $\mathcal{F}(d^\pi)$ of the induced state-distribution $d^\pi$. Crucially, General Utilities (or Convex) RL, which we already discuss within the Related Works section, is a sub-field of RL theory without any significant real-world application to date. Indeed, our work aims to leverage understanding from this field to control generative models in ways that go beyond the reach of standard RL formulations, which currently are the standard way for fine-tuning of generative models. In particular, this work is the first establishing a connection between (reward-guided) flow model merging and General Utilities (or Convex) RL. Crucially, we believe that this is a positive contribution since General Utilities RL brings an algorithmic machinery strictly more general than RL (although here it is sufficient to consider a sub-case which collapses to standard calculus of variations), while its methods usually do not scale well to high-dimensional settings due to the inability of representing complex distributions, which is here solved by leveraging diffusion/flow models.

---

> > > ### Author Response · Authors · 2025-11-23
> > >
> > > **Baselines**
> > >
> > > As clarified within the previous points, our work neither tackles inference-time model composition, nor classic fine-tuning, and it is to our knowledge the first introducing the  problem of reward-guided flow merging. As a consequence, comparing against "baselines on composition and fine-tuning" is impossible, as these (model composition and fine-tuning) are other problems than the one treated within this work. Due to the lack of existing baselines, within the Experiments section we have provided both illustrative, visually interpretable settings (see Fig. 2 and 3) to showcase the ability of the proposed algorithm to solve diverse (reward-guided) model merging tasks, as well as evaluations on higher-dimensional tasks (see Fig. 3,4,5). In particular, while within the previous version of the paper evaluation was performed only on one real-world molecular design task, we have updated the manuscript with another evaluation on a conformer generation task, presented within the paragraph "Flow Merging of Conformer Generation Models" in Sec. 7. The presented experimental evaluation shows that RFM can convincingly tackle the presented reward-guided flow merging problem both in settings with synthetic and real-world data. To our knowledge, this is the first method that achieves so (both theoretically and practically).
> > >
> > > **Computational Cost and Approximate Oracle**
> > >
> > > We thank the Reviewer for this interesting question. We have updated the paper with a new Appendix F entitled "Computational Complexity, Cost, and Approximate Fine-Tuning Oracles", where we provide a discussion of computational cost, complexity, and usage of RFM with an approximate fine-tuning oracle. In short, a theoretical computational complexity analysis would lead to a linear dependency on $K$, multiplied by the fine-tuning oracle cost. In practice, we show experimentally that the number of iterations $K$ can be traded-off with the quality of the fine-tuning oracle, which can be expressed by $N$ in our case (see Appendix E), leading to nearly-identically optimal results by using approximate oracles  (i.e. lower $N$) for a higher number of iterations (i.e. higher $K$), or better oracles (i.e. higher $N$) for a lower number of iterations (i.e. lower $K$). This viewpoint is motivated by standard understanding in convex optimization (see Appendix F).
> > >
> > > **References**
> > >
> > > [14] Diffusion soup: Model merging for text-to-image diffusion models, B. Biggs, 2024.
> > >
> > > [15] Fine-tuning of diffusion models via stochastic control: entropy regularization and beyond, W. Tang., 2024.
> > >
> > > [16] Large Language Diffusion Models, Shen Nie, 2025.
> > >
> > > [17] Flow Density Control: Generative Optimization Beyond Entropy-Regularized Fine-Tuning, R. De Santi, 2025.

---

### Author Response · Authors · 2025-11-23
**Global Comment**

We thank the Reviewers for recognizing our framework as unifying two difficult problems in AI for science (Reviewer sG1a), novel and well-motivated (Reviewer vWQC), theoretically sound (Reviewers sG1a and vWQC), promising (Reviewer vWQC), and considering the proposed method well-articulated and illustrated (Reviewer vWQC). We have improved the paper by using one extra page (blue text), according to ICLR guidelines, to address the Reviewers constructive feedback. Among other things, within the updated version of the manuscript, we have significantly expanded and strengthened the experimental evaluation of the proposed method via the following new contributions:


**New experimental case study of RFM for molecular conformer generation via ETFlow**

We added a new experimental case study of RFM on a molecular conformer generation task based on the recent ETFlow model [1]. This is reported within the added paragraph "Flow Merging of Conformer Generation Models" (page 9), with results reported in Fig. 5, and experimental details in Appendix F.3.

**Strengthened evaluation of reward-guided flow merging**

As mentioned by Reviewer vWQC, the evaluation of reward-guided flow merging on real-world experiments within the previous version of the work was reported only within the appendix and not convincing for the molecular design task. We have updated the manuscript with significantly stronger results for reward-guided flow merging (beyond pure flow merging) for both 3D molecular design and the newly added conformer generation tasks.

We believe that the updated experimental section showcases significantly more convincing, strong, and promising performance for a fundamentally new, theory-backed method (i.e., RFM), that we view as the central contribution of this work, which is primarily of mathematical and algorithmic nature.

--- (added later) ---

**Evaluation on image domain**

As further mentioned by Reviewer vWQC, we have also performed a small-scale evaluation on another domain, namely image generation. We report an overview and results within Appendix H (page 27, orange text) titled "Beyond Molecules: Reward-Guided Flow Merging of Pre-Trained Image Models". To summarize, we show that RFM can successfully perform reward-guided, i.e., maximize the aesthetic score of the fine-tuned model while preserving information from multiple prior models.

**References**

[1] Et-flow: Equivariant flow-matching for molecular conformer generation, Hassan Majdi, 2024.

---

### Meta-Review · Area_Chair_JsfN · 2026-01-05

**Summary:**

Strength: The paper proposes a unified framework for generative optimization. It then developed the reward-guided flow-merging algorithm. Theoretical results are provided and experimental results show the performance of the algorithm.

Weakness: The novelty of theoretical results/technique should be better highlighted. Also, reviewers have raised concerns regarding the baseline comparison and application scenarios.

**Reviewer Concerns:**

Weakness: The novelty of theoretical results/technique should be better highlighted. Also, reviewers have raised concerns regarding the baseline comparison and application scenarios.

**Reviewer Scores:**

Based on the comments and discussions, the reviewers will likely keep their scores.

---

### Decision · Program_Chairs · 2026-01-26

Reject